# mGluR5-mediated astrocytes hyperactivity in the anterior cingulate cortex contributes to neuropathic pain in male mice

Weida Shen [1,2,6] ✉, Fujian Chen[1,6], Yejiao Tang[2,3,6], Yulu Zhao[2,6], Linjing Zhu[2,6], Liyang Xiang[4], Li Ning[5], Wen Zhou[2], Yiran Chen[2], Liangxue Wang[1], Jing Li[1], Hui Huang[1] & Ling-Hui Zeng [1,2] ✉

Astrocytes regulate synaptic transmission in healthy and pathological conditions, but their involvement in modulating synaptic transmission in chronic pain is unknown. Our study demonstrates that astrocytes in the anterior cingulate cortex (ACC) exhibit abnormal calcium signals and induce the release of glutamate in male mice. This leads to an elevation in extracellular glutamate concentration, activation of presynaptic kainate receptors, and an increase in synaptic transmission following neuropathic pain. We discovered that the abnormal calcium signals are caused by the reappearance of metabotropic glutamate receptor type 5 (mGluR5) in astrocytes in male mice. Importantly, when we specifically inhibit the Gq pathway using iβARK and reduce the expression of mGluR5 in astrocytes through shRNA, we observe a restoration of astrocytic calcium activity, normalization of synaptic transmission and extracellular concentration of glutamate, and improvement in mechanical allodynia in male mice. Furthermore, the activation of astrocytes through chemogenetics results in an overabundance of excitatory synaptic transmission, exacerbating mechanical allodynia in mice with neuropathic pain, but not in sham-operated male mice. In summary, our findings suggest that the abnormal calcium signaling in astrocytes, mediated by mGluR5, plays a crucial role in enhancing synaptic transmission in ACC and contributing to mechanical allodynia in male mice.

The anterior cingulate cortex (ACC) plays vital roles in learning, memory, attention, as well as in pain perception and processing[1–3]. Recent studies employing various experimental techniques have consistently demonstrated crucial functions of the ACC in the processing of pain-related information in humans and in the behavioral responses of animals to noxious stimuli or tissue injury[4,5]. Cumulative evidence demonstrates that chronic pain-induced changes in ACC affect synaptic transmission[1,6–12]. However, the exact mechanisms underlying the alteration of synaptic transmission in pain processing remain unclear.

Astrocytes are the most abundant type of glial cell in the central nervous system (CNS)[13]. Astrocytes not only provide energy substrates, but also play a crucial role in supporting synaptic function and plasticity, as well as regulating regional cerebral blood flow (CBF)[14–18]. The alteration of glial cell function is a common hallmark of many diseases in the CNS[19–24]. Studies have indicated that inflammation and nerve injury in the periphery can activate spinal astrocytes[25,26], and inhibiting astroglial function in the spinal cord has been shown to alleviate mechanical allodynia[27,28]. Moreover, recent findings clearly show that astrocytes in the primary somatosensory (S1) cortex play an essential role in the induction of mechanical allodynia[29–31].

Accumulating evidence suggests that intracellular calcium is a vital signaling pathway that regulates the release of gliotransmitters and modulates synaptic transmission[32–34]. Studies have demonstrated abnormal $Ca^{2+}$

[1]Anji People's Hospital, Affiliated Anji Hospital, School of Medicine, Hangzhou City University, Hangzhou, China. [2]Key Laboratory of Novel Targets and Drug Study for Neural Repair of Zhejiang Province, School of Medicine, Hangzhou City University, Hangzhou, China. [3]Institute of Pharmacology & Toxicology, College of Pharmaceutical Sciences, Key Laboratory of Medical Neurobiology of the Ministry of Health of China, Zhejiang University, Hangzhou, China. [4]School of Medicine, Nankai University, Tianjin, China. [5]Department of Anesthesiology and Surgical Intensive Care Unit, Xinhua Hospital, Shanghai Jiaotong University School of Medicine, Shanghai, China. [6]These authors contributed equally: Weida Shen, Fujian Chen, Yejiao Tang, Yulu Zhao, Linjing Zhu. ✉e-mail: shenwd@hzcu.edu.cn; zenglh@hzcu.edu.cn

activities in astrocytes during various pathological conditions, such as Alzheimer's disease[35–38], epilepsy[39,40], and neuropathic pain[41,42]. Additionally, metabotropic glutamate receptor type 5 (mGluR5) expression, which plays a key role in mediating $Ca^{2+}$ signals in astrocytes, has been shown to reemerge in adult astrocytes during certain pathological conditions[43–45], such as neuropathic pain[31]. However, it is currently unclear how astrocyte $Ca^{2+}$ signals in the ACC are altered following chronic pain and whether these changes contribute to the modulation of synaptic transmission and the development of mechanical allodynia.

In this study, we aimed to elucidate the role of astrocytes in modulating synaptic transmission in ACC after peripheral nerve injury. We found that mGluR5 mediates astrocytic hyperactivity following chronic pain. Moreover, we observed that chronic pain is associated with an increase in both the frequency and amplitude of mEPSCs in the ACC in male mice. Additionally, our microdialysis and electrophysiological experiments revealed that chronic pain induces an elevation in extracellular glutamate levels within the ACC region, thereby enhancing synaptic transmission through the activation of presynaptic receptors in male mice. Selective inhibition of Gq signaling using iβARK in astrocytes rescues aberrant calcium signals, decreases synaptic transmission and extracellular concentration of glutamate, and alleviates mechanical allodynia in male mice. Additionally, the direct and selective activation of astrocytes with Gq designer receptors exclusively activated by designer drugs (DREADDs) leads to increased synaptic transmission in vitro in male mice. However, this manipulation does not induce mechanical allodynia in sham-operated mice in vivo. On the other hand, CNO administration exacerbates mechanical allodynia in nerve-injured male mice. Furthermore, by conditionally reducing the expression of mGluR5 in astrocytes of the ACC in male mice, we observed the normalization of calcium signals, synaptic transmission, extracellular glutamate concentration, and the alleviation of mechanical allodynia in a chronic pain model induced by nerve injury. Taken together, these results are the first to indicate that astrocytic hyperactivity, mediated by astrocytic mGluR5, is an integral component of a form of synaptic plasticity in the ACC following chronic pain in male mice. The study of mGluR5-mediated signaling pathway in astrocytes may provide novel targets for the future treatment of chronic pain.

## Results

### Chronic pain-induced astrocytes exhibit hyperactivity $Ca^{2+}$ signaling and increase extracellular glutamate concentration in ACC in male mice

In this study, GCaMP7b was utilized to observe astrocyte $Ca^{2+}$ signaling in the anterior cingulate cortex (ACC) in male mice. To deliver GCaMP7b to astrocytes in vivo, we used local microinjections of adeno-associated viruses (AAV2/5) with an astrocyte-specific GfaABC1D promoter. Consistent with previous work using AAV2/5 together with GfaABC1D promoter in hippocampus[46,47], our results demonstrated that within the virally transduced region, GCaMP7b was expressed in astrocytes in cortex with high penetrance (85.09 ± 2.637% of GFAP-positive cells expressed GCaMP7b, from 4 mice) (Fig. 1a, b). No co-staining was observed with the neuronal marker, NeuN (Fig. 1b). Furthermore, additional staining for markers of other glial cell types, including Iba1 for microglia and NG2 for oligodendrocyte precursor cells (OPCs), confirmed that GCaMP7b expression was not observed in microglia (0 ± 0% of Iba1-positive cells expressed GCaMP7b) or OPCs (0 ± 0% of NG2-positive cells expressed GCaMP7b) (Fig. 1b). Thus, the data support the conclusion that GCaMP7b is selectively expressed in GFAP-expressing astrocytes in the cortex, without expression in neurons, microglia, or OPCs.

Accumulating evidence indicates that astrocytes display aberrant $Ca^{2+}$ signaling in pathological conditions[35–42]. In light of this, we first explored whether chronic pain affected spontaneous $Ca^{2+}$ signals in ACC slices under conditions where neuronal activity was pharmacologically blocked by TTX (1 μM) and picrotoxin (100 μM). In accordance with previous work on S1 astrocytes following peripheral nerve injury[30,31], we found that chronic pain led to a statistically larger increase of $Ca^{2+}$ signal frequency and amplitude

than the sham group in male mice (Fig. 1c–e). Recent evidence indicates that with neuropathic pain, synaptic transmission in ACC is modulated, which is characterized by enhanced excitatory synaptic transmission[7,12,48]. To investigate whether synaptic transmission in ACC pyramidal neurons was altered after peripheral nerve injury in our experimental conditions, we recorded miniature excitatory postsynaptic currents (mEPSCs) in layer II/III pyramidal neurons. Consistent with previous studies[48], we observed a significant increase in both mEPSC frequency and amplitude in ACC neurons after peripheral nerve injury, compared with the control group in male mice (Fig. 2a–e). To investigate whether the enhanced excitatory synaptic transmission in the ACC after peripheral nerve injury is mediated by presynaptic or postsynaptic mechanisms, we examined the paired-pulse ratio (PPR) in ACC neurons. Our findings revealed that the increase in mEPSC frequency induced by peripheral nerve injury was associated with a decrease in PPR in male mice (Fig. 2f, g), indicating the presence of a presynaptic mechanism. To further confirm the presence of chronic pain in these animals, we analyzed the mechanical thresholds for the mice used in calcium imaging and in electrophysiology, and found significant reductions in mechanical thresholds in CCI male mice compared to sham controls (CCI, $n = 12$ mice, paw withdrawal threshold = 0.7244 ± 0.05726 g; sham, $n = 13$ mice, paw withdrawal threshold=1.456 ± 0.08568 g; $p < 0.0001$, unpaired t-test).

Previous research has indicated that chronic pain induces presynaptic long-term potentiation (pre-LTP) via GluK1-containing kainate receptors[49]. However, the source of glutamate that activates presynaptic kainate receptors is still unknown. Conversely, multiple studies have demonstrated that astrocytic calcium signaling stimulates the release of glutamate[50–54]. Thus, we measured the extracellular concentration of glutamate using microdialysis and found that chronic pain led to a significant increase in its levels in male mice (Fig. 2h–i). These results suggest that abnormal calcium signaling in astrocytes may be associated with the release of glutamate and the subsequent increase in extracellular concentration in male mice.

### mGluR5 reemerged in ACC astrocytes following chronic pain in male mice

Astrocytic mGluR5 exhibits a high level of expression in immature astrocytes, which leads to the formation of robust $Ca^{2+}$ signals. However, during postnatal development, its expression and function decrease dramatically and eventually disappear within a few weeks after birth[55,56]. Nonetheless, it has been observed that astrocytic mGluR5 can reappear in the adult brain during certain pathological conditions[31,57,58]. For instance, an elegant study by Koizumi and his colleagues demonstrated that mGluR5 reemerged in astrocytes of the S1 region following peripheral nerve injury[29–31]. We investigated whether peripheral nerve injury caused the re-expression of mGluR5 in the ACC. Our observations indicated that astrocytic mGluR5 was significantly expressed on the 14th days after peripheral nerve injury (Fig. 3a). In contrast, we did not find any expression of mGluR5 in astrocytes of control male mice that underwent sham surgery (Fig. 3a). Moreover, we observed mGluR5 expression in astrocytes on the contralateral side of the CCI. However, on the ipsilateral side of the CCI, no mGluR5 expression was detected in astrocytes, despite using the same antibody on the same slice (Supplementary Fig. 1). Immunohistochemistry revealed a significant upregulation of mGluR5 in astrocytes of CCI male mice, as evidenced by the increased presence of mGluR5 (Fig. 3b).

To determine whether astrocytic mGluR5 in the ACC contribute to the increase of $Ca^{2+}$ signaling in astrocytes, $Ca^{2+}$ signals were monitored in the presence of the specific mGluR5 antagonist MPEP. To avoid MPEP affecting neuronal excitability[59], the experiments were conducted under conditions where the neuronal network was blocked by the addition of TTX and picrotoxin. We observed that the drug significantly reduced the frequency and amplitude of astrocytic transients in the ACC astrocytes of male mice with peripheral nerve injuries (Fig. 3c–e). To further validate our findings, we tested the response to the mGluR5 agonist CHPG (500 μM) in both sham and CCI conditions, in the presence of TTX and picrotoxin. In sham conditions, CHPG did not evoke calcium transients in astrocytes

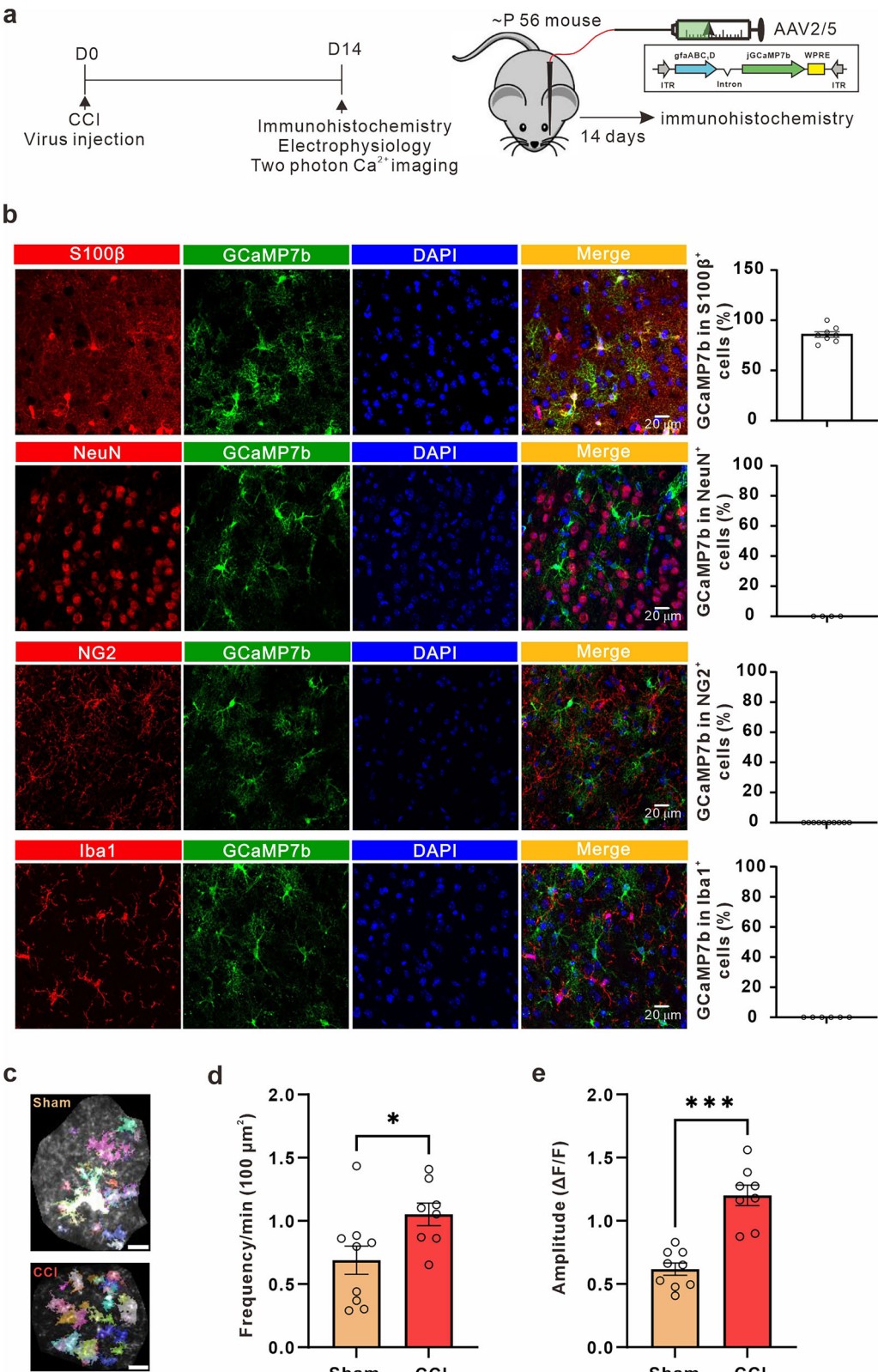

**Fig. 1 | Neuropathic pain induces astrocyte hyperactivity in ACC. a** Schematic representation of the experimental design. **b** Illustration detailing the co-localization of jGCaMP7b (green) with the astrocytic-specific marker s100β (red) in the ACC (85.75 ± 2.75%, *n* = 8 slices from 3 mice), without overlap with the neuronal marker NeuN (red) (0.00 ± 0.00%, *n* = 4 slices from 2 mice), Microglia specific marker Iba1 (red) (0.00 ± 0.00%, *n* = 10 slices from 3 mice) and oligodendrocyte precursor cell specific marker NG2 (red) (0.00 ± 0.00%, *n* = 6 slices from 3 mice). **c** Representative images displaying all AQuA-detected events from a 1-minute ex vivo astrocytic GCaMP7b $Ca^{2+}$ imaging experiment. Each color represents an individual event and

is randomly assigned. Scale bar: 10 μm. **d** Graph depicting the frequency of astrocytic $Ca^{2+}$ signals in the ACC of sham and CCI mice (*n* = 9 cells from 4 sham mice; *n* = 8 cells from 4 CCI mice; sham (yellow) = 0.6897 ± 0.1250 min/100 μm², CCI (red) = 1.052 ± 0.08905 min/100 μm², *p* = 0.0359, unpaired t-test). **e** Graph showing the amplitude of astrocytic $Ca^{2+}$ signals in the ACC of sham and CCI mice (*n* = 9 cells from 4 sham mice; *n* = 8 cells from 4 mice CCI; sham (yellow) = 0.6172 ± 0.04862 ΔF/F, CCI (red) = 1.202 ± 0.0816 ΔF/F, *p* < 0.0001, unpaired t-test). Error bars represent the mean ± SEM.

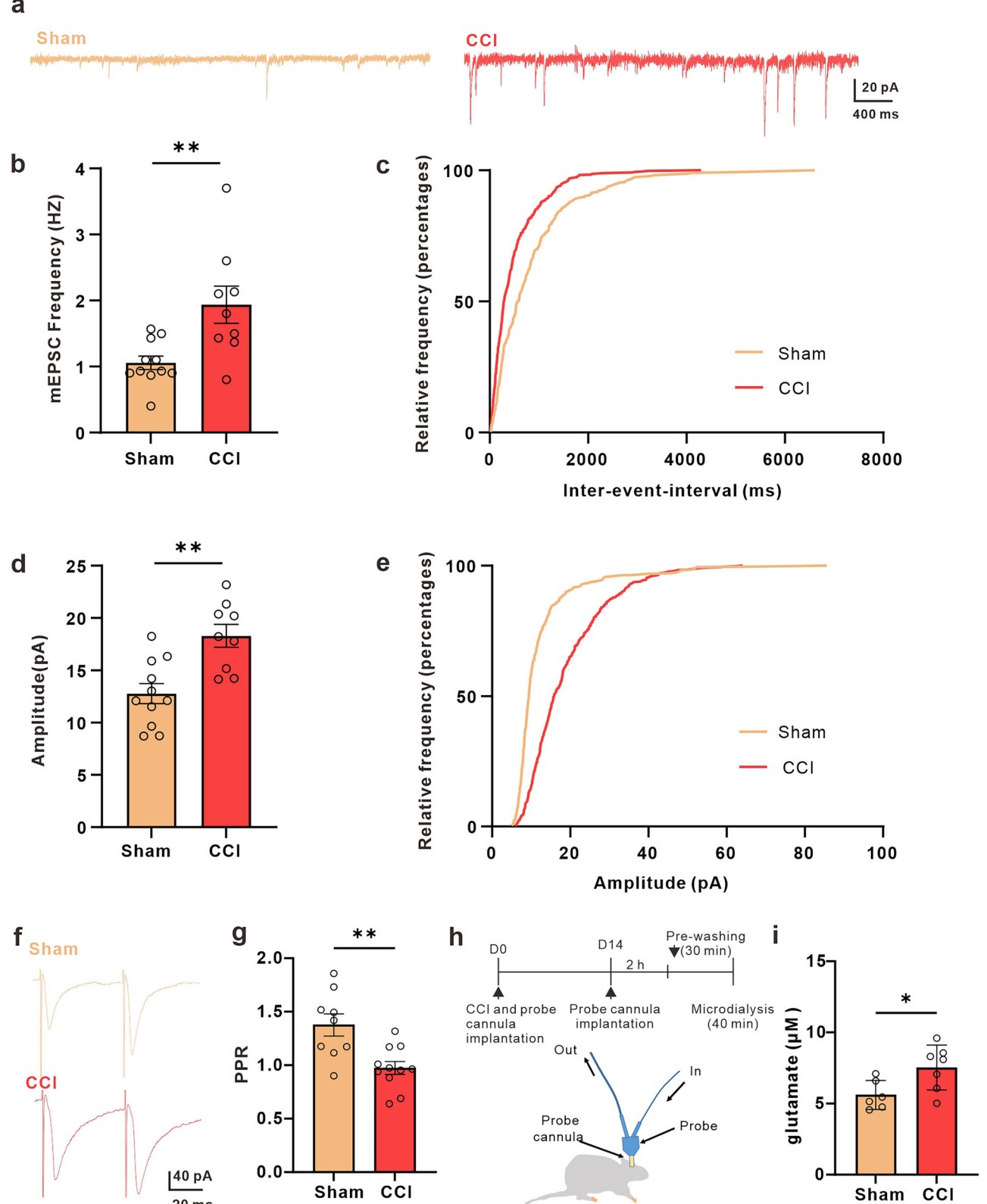

(Supplementary Fig. 2 a, b, e). However, in CCI conditions, CHPG significantly induced calcium transients in astrocytes (Supplementary Fig. 2 c, d, f). Moreover, the $Ca^{2+}$ signals elicited by CHPG were attenuated in the presence of the mGluR5 antagonist MPEP (Supplementary Fig. 2g). These data suggest that astrocytic hyperactivity is caused by the reemergence of mGluR5 in ACC astrocytes in male mice.

**Reducing ACC astrocyte Gq-dependent signaling rescues chronic pain-induced symptoms in male mice**

Our findings indicate that peripheral nerve injury leads to changes in both synaptic transmission and astrocytic $Ca^{2+}$ signaling. However, the specific connection between abnormal $Ca^{2+}$ signaling and enhanced synaptic transmission is still not well understood. To investigate whether aberrant

**Fig. 2 | Neuropathic pain increases extracellular glutamate concentration and synaptic transmission in ACC. a** Representative traces of miniature excitatory postsynaptic currents (mEPSCs) recorded from layer II-III neurons of the ACC in sham (yellow) and CCI (red) mice. Summary mEPSCs frequency (**b**, $n = 11$ cells from 5 sham mice; $n = 9$ cells from 4 CCI mice; sham (yellow) = $1.058 \pm 0.1022$ Hz, CCI (red) = $1.937 \pm 0.2812$ Hz, $p = 0.0053$, unpaired t-test) and amplitude (**d**, sham (yellow) = $12.78 \pm 0.9515$ pA, CCI (red) = $18.30 \pm 1.083$ pA, $p = 0.0012$, unpaired t-test) in sham and CCI mice. Cumulative probability plots of mEPSCs inter-event intervals (IEI) (**c**, $p < 0.0001$, Kolmogorov-Smirnov Test) and mEPSCs amplitude (**e**, $p < 0.0001$, Kolmogorov-Smirnov Test) in sham (yellow) and CCI (red) mice.

**f** Representative eEPSC traces evoked by pair pulse protocol (PPP) with an interval of 50 ms recorded in the ACC in sham (yellow) and CCI (red) mice. **g** Summary bar graph of paired-pulse ratio (PPR) index ($n = 9$ cells from 4 sham mice; $n = 11$ cells from 4 CCI mice; sham (yellow) = $1.376 \pm 0.1033$, CCI (red) = $0.9743 \pm 0.06052$, $p = 0.0025$, unpaired t-test) of eEPSCs in sham and CCI mice. **h** Schematic of the experimental design of in vivo microdialysis. **i** Extracellular glutamate concentrations in the dialysate in the ACC of sham and CCI mice ($n = 6$ sham mice; $n = 7$ CCI mice; sham (yellow) = $5.598 \pm 0.4172$ μM, CCI (red) = $7.529 \pm 0.5957$ μM, $p = 0.0263$, unpaired t-test). Error bars represent the mean ± SEM.

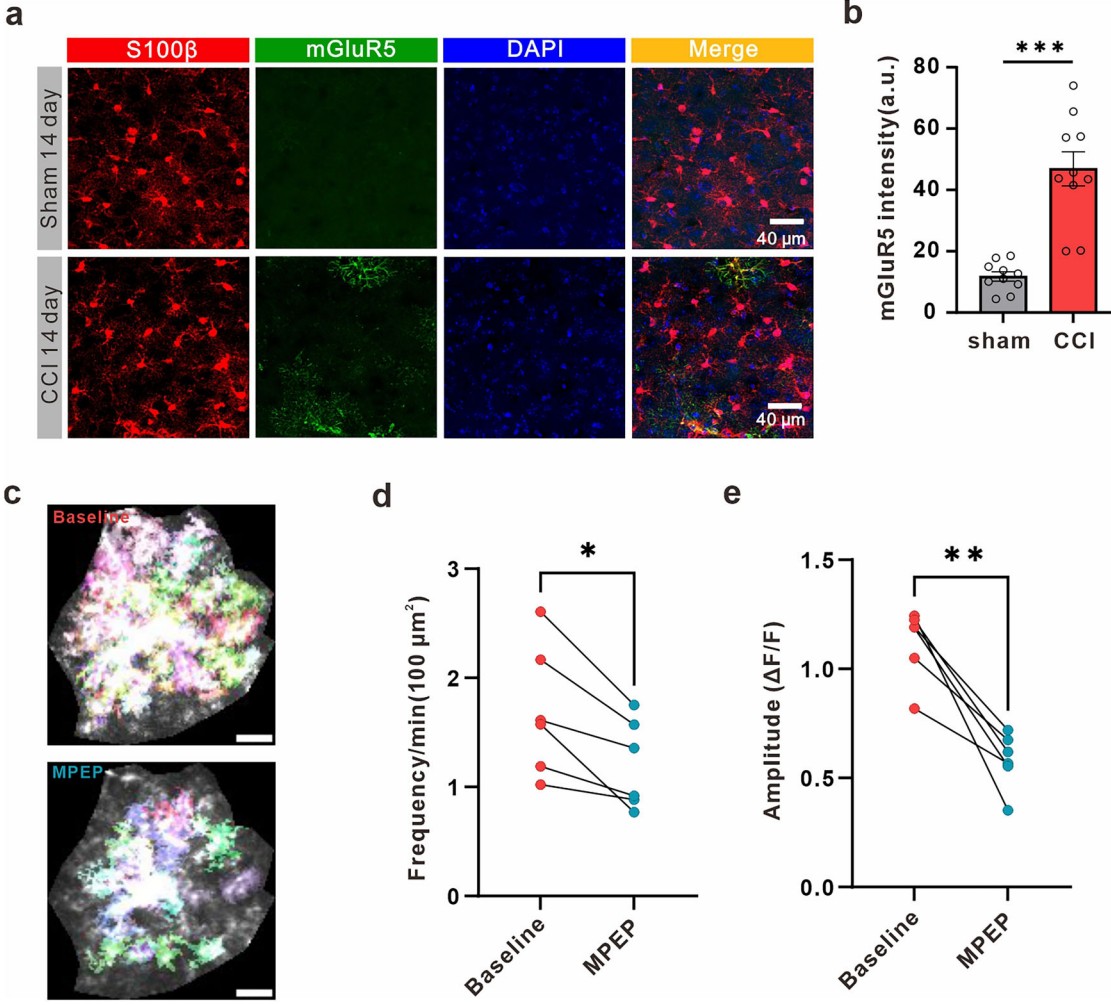

**Fig. 3 | Reemergence of mGluR5 in ACC astrocytes after neuropathic pain responsible for promoting astrocytic Ca²⁺ signals. a** Immunohistochemical images of astrocytes in the ACC of sham or CCI mice. Red indicates s100β, green indicates mGluR5, and blue indicates nuclei. **b** Quantification of mGluR5 fluorescence intensity in ACC astrocytes from sham and CCI mice ($n = 10$ slices from 5 sham mice, $n = 10$ slices from 5 CCI mice, sham (gray) = $11.7 \pm 1.507$ a.u., CCI (red) = $46.89 \pm 5.56$ a.u., $p < 0.0001$, un paired t-test). **c** Representative images displaying all AQuA-detected events from a 1-minute ex vivo astrocytic GCaMP7b Ca²⁺ imaging experiment. Upper: representative image before application of the mGluR5 antagonist MPEP; below: representative image after application of MPEP.

Colors indicate detected events. Scale bar: 10 μm. **d** Graph depicting the frequency of astrocytic Ca²⁺ signals in the ACC of CCI mice before and after application of the mGluR5 antagonist MPEP ($n = 6$ cells from 3 mice; baseline (red) = $1.695 \pm 0.2438$ min/100 μm², MPEP (blue) = $1.209 \pm 0.1665$ min/100 μm², $p = 0.0117$, paired t-test). **e** Graph showing the amplitude of astrocytic Ca²⁺ signals in the ACC of CCI mice before and after application of the mGluR5 antagonist MPEP ($n = 6$ cells from 3 mice; baseline (red) = $1.119 \pm 0.06624$ ΔF/F, MPEP (blue) = $0.5810 \pm 0.05255$ ΔF/F, $p = 0.0020$, paired t-test). Error bars represent the mean ± SEM.

Ca²⁺ signaling contributes to enhanced synaptic transmission, we attempted to selectively inhibit the Gq-signaling pathway in astrocytes using iβARK[60]. To validate the inhibitory effect of iβARK on astrocytic Gq pathway-induced Ca²⁺ transients, we co-expressed iβARK and GCaMP7b (Supplementary Fig. 3a, b). Quantitative analysis revealed that within the virally transduced region, iβARK (mCherry) expression was largely confined to astrocytes, with

high penetrance ($78.18 \pm 1.975\%$ of S100β-positive cells expressed mCherry, from 3 mice). In mice co-injected with iβARK (mCherry) and GCaMP7b, GCaMP7b expression was predominantly observed in iβARK (mCherry) positive cells, also with high penetrance ($94.27 \pm 3.198\%$ of mCherry-positive cells expressed GCaMP7b, from 3 mice). Additional staining for markers of other glial cell types confirmed that iβARK expression was absent

in microglia (0 ± 0% of Iba1-positive cells expressed mCherry) and OPCs (0 ± 0% of NG2-positive cells expressed mCherry).

Subsequently, we evaluated the impact of iβARK expression on mGluR5 agonist-evoked responses in astrocytes from the ACC of male mice with peripheral nerve injuries. The slices were perfused with 1 μM tetrodotoxin (TTX) and 100 μM picrotoxin, and the astrocytes were stimulated using CHPG (500 μM), a specific agonist targeting the mGluR5 subtype. This compound effectively triggered $Ca^{2+}$ increases in ACC astrocytes that expressed only mCherry in male mice with peripheral nerve injuries (Supplementary Fig. 4a–c). However, astrocytes expressing iβARK exhibited almost complete suppression of CHPG-evoked responses (Supplementary Fig. 4d–f). The results suggest that the approach of inhibiting the Gq-signaling pathway has been effective. Next, we conducted experiments to investigate whether iβARK can inhibit astrocytic hyperactivity caused by peripheral nerve injury. In male mice with peripheral nerve injury, iβARK significantly reduced the frequency and amplitude of $Ca^{2+}$ signaling in astrocytes of the ACC (Fig. 4a–c). However, mice expressing only mCherry astrocytes still showed abnormal $Ca^{2+}$ signaling (Fig. 4a–c). We then investigated whether iβARK could alleviate increased excitatory synaptic transmission and mechanical allodynia. We found that the amplitude and frequency of mEPSC were decreased (Fig. 4d–h), and the PPR is increased in the ACC of male mice expressing iβARK compared to CCI male mice (Fig. 4i, j). Furthermore, we observed that the expression of iβARK in astrocytes was associated with a reduction in extracellular glutamate concentration in the ACC of male mice with peripheral nerve injury (Fig. 4k), which correlated with a reduction in mechanical allodynia (Fig. 4l). In contrast, male mice expressing only mCherry showed continued abnormal synaptic transmission and higher extracellular glutamate concentration in the ACC, which was associated with persistent mechanical allodynia when compared to the iβARK group (Fig. 4d–l).

## Enhancing astrocytic $Ca^{2+}$ signaling in the ACC astrocytes aggravates mechanical allodynia in nerve-injured mice, but not in sham-operated male mice

To further confirm the role of astrocytic $Ca^{2+}$ signaling in neuropathic pain, we expressed hM3Dq to stimulate the Gq signaling pathway and examined the resulting effects on synaptic transmission and mechanical allodynia. First, we confirmed that hM3Dq was selectively expressed in astrocytes (Supplementary Fig. 5a, b). By bath application of CNO, we observed an increase in $Ca^{2+}$ signals in astrocytes co-expressing hM3Dq (red) and GCaMP7b (green), providing additional confirmation that chemogenetics can enhance astrocytic $Ca^{2+}$ signaling in sham male mice (Supplementary Fig. 5c-d). To confirm that the effects of CNO are mediated via Designer Receptors Activated Only by Designer Drugs (DREADDs), we tested CNO on sham mice expressing only mCherry. Two photon $Ca^{2+}$ imaging showed that CNO did not evoke calcium transients in astrocytes (Supplementary Fig. 5d). Next, we measured mEPSCs in ACC neurons before and after a 5 min bath application of CNO. We found that while CNO increased the frequency of mEPSCs, it did not affect their amplitude (Fig. 5a–e). The increased frequency of mEPSC was associated with decreased PPR, suggesting the presence of a presynaptic mechanism (Fig. 5f, g). Additionally, we administered a single dose of 1 mg/kg CNO (i.p.) 1 hour before behavioral analysis to test for sensitization to noxious stimuli, but found no significant difference between before and after CNO administration in sham male mice (Fig. 5h–i). Next, we examined whether activation of astrocytes in the ACC of mice with peripheral nerve injury could affect their response to noxious stimuli. We found that CNO aggravates responses to noxious stimuli in the hM3Dq group when compared to before CNO treatment (Fig. 5j).

These results indicate that increasing ACC astrocyte Gq pathway signaling does not affect responses to noxious stimuli in sham male mice. However, stimulating Gq pathway signaling in nerve-injured male mice worsens mechanical allodynia.

## Knockdown of mGluR5 in astrocytes attenuated chronic pain in male mice

Our data strongly suggest that the re-emergence of mGluR5 in ACC astrocytes contributes to the enhancement of excitatory synaptic transmission and neuropathic mechanical allodynia. Based on these findings, our aim was to investigate whether viral delivery of mGluR5-shRNA, specifically targeting the knockdown of mGluR5 expression in astrocytes, could reverse the changes observed in mice with neuropathic pain induced by peripheral nerve injury. Immunohistochemistry performed after 3 weeks of injection revealed mCherry expression in S100β-positive astrocytes (Fig. 6a, b), indicating successful AAV2/5-mediated viral transduction. The group receiving mGluR5-shRNA delivery showed significantly reduced levels of mGluR5 expression in astrocytes compared to the group expressing with scramble-shRNA (Fig. 6b2). To confirm that mGluR5 is effectively knocked out in astrocytes, we stimulated these cells with an mGluR5 agonist in both the mGluR5-shRNA and scramble-shRNA groups. To visualize astrocytic $Ca^{2+}$ signals, we injected a mixture of mGluR5-shRNA-mCherry and GCaMP7b. Consistent with the immunohistochemistry results, we observed that the mGluR5-shRNA group exhibited significantly reduced evoked $Ca^{2+}$ signals compared to the scramble-shRNA group (Fig. 6c).

Furthermore, we found that shRNA significantly decreases the frequency and amplitude of $Ca^{2+}$ fluctuations in ACC astrocytes of peripheral nerve-injured male mice (Fig. 6d–f). When astrocytes expressed shRNA-mCherry targeting mGluR5 for 21 days, the amplitude and frequency of mEPSCs in ACC neurons (Fig. 7a–g), as well as the extracellular glutamate concentration in ACC (Fig. 7h), were significantly decreased compared to CCI male mice. Additionally, we found that the mechanical allodynia caused by peripheral nerve injury was reduced by mGluR5-shRNA (Fig. 7i). Overall, our findings suggest that the reemergence of astrocytic mGluR5 may be a critical event that controls synapse plasticity in the ACC and contribute to the development of mechanical allodynia in male mice.

## Discussion

In the present study, we have demonstrated that the re-emergence of mGluR5 in ACC astrocytes is responsible for the hyperactivity of astrocytic calcium signaling, leading to an increase in synaptic transmission in ACC and mechanical allodynia in male mice. Furthermore, we showed that activation of astrocytes with chemogenetics in ACC in vivo aggravates mechanical allodynia in chronic pain conditions in male mice. Additionally, our findings revealed that the knockdown of mGluR5 in astrocytes of the ACC in male mice with nerve injuries resulted in the alleviation of mechanical allodynia. This is the first study, to our knowledge, to reveal the reemergence of mGluR5 expression in astrocytes of animals experiencing chronic pain, along with alterations in aberrant astrocytic $Ca^{2+}$ signaling and synaptic plasticity within the ACC in male mice.

Several studies have shown that chronic pain induces alterations in synaptic transmission[8,11,12,61,62]. However, the underlying mechanisms remain unclear. Recently, a remarkable study by Danjo demonstrated that activation of mGluR5 enhances $Ca^{2+}$ signaling in S1 astrocytes, subsequently triggering the expression of several synaptogenic molecules such as TSP1, Glypican-4, and Hevin[31]. These molecular changes contribute to the formation of an excessive number of excitatory synapses and persistent modifications in neuronal activity within the S1 cortex. Consequently, this process ultimately culminates in the development of refractory neuropathic pain. In our study, we observed a difference in the time course of mGluR5 reemergence compared to a recent report on the S1 cortex. In the S1 cortex, astrocytic mGluR5 signals remained sustained for 7 days and then returned to the control level by day 10[31]. This transient increase suggests that astrocytic mGluR5 plays a role in the development of neuropathic pain. Differences between models and brain regions might account for this discrepancy. Indeed, in our experimental model, we applied three loosely constrictive ligatures (6-0 suture) around the sciatic nerve with a spacing of 1-1.5 mm between each ligature. In contrast, Koizumi and colleagues utilized an 8-0 suture to ligate one-third to one-half of the right sciatic nerve. Our data align

**Fig. 4 | iβARK decreases astrocytic Ca²⁺ signals, synaptic transmission, extracellular concentration of glutamate in ACC, and alleviates mechanical allodynia in neuropathic pain mice.**

**a** Representative images displaying all AQuA-detected events from a 1-minute ex vivo astrocytic GCaMP7b Ca²⁺ imaging experiment from different groups. Upper: representative image in CCI mice; middle: representative image in iβARK-expression mice; below: representative imaging in mCherry-expression mice; Colors indicate detected events. Scale bar: 10 μm. **b** Graph depicting the frequency of astrocytic Ca²⁺ signals in the ACC of CCI, iβARK and mCherry mice ($n = 6$ cells from 3 CCI mice; $n = 9$ cells from 4 iβARK mice; $n = 6$ cells from 3 mCherry mice; CCI (red) = 1.169 ± 0.1352 min/100 μm², iβARK (blue) = 0.7121 ±

0.08085 min/100 μm², mCherry (pink) = 1.109 ± 0.1086 min/100 μm², $p = 0.0088$, F = 6.221, one-way ANOVA followed by Dunnett's test). **c** Graph showing the amplitude of astrocytic Ca²⁺ signals in the ACC of CCI, iβARK and mCherry mice (CCI (red) = 1.367 ± 0.2643 ΔF/F, iβARK (blue) = 0.7574 ± 0.04621 ΔF/F, mCherry (pink) = 1.150 ± 0.09837 ΔF/F, $p = 0.0179$, F = 5.068, one-way ANOVA followed by Dunnett's test. **d** Representative traces of miniature excitatory postsynaptic currents (mEPSCs) recorded from layer II-III neurons of ACC in CCI (red), iβARK (blue) and mCherry (pink) mice. Summary of mEPSCs frequency (**e**, $n = 8$ cells from 3 CCI mice; $n = 10$ cells from 4 iβARK mice; $n = 5$ cells from 3 mCherry mice; CCI (red) = 2.317 ± 0.2897 Hz, iβAKR (blue) = 0.9633 ± 0.06506 Hz, mCherry (pink) = 2.067 ± 0.2584 Hz, $p = 0.0001$, F = 14.22, one-way ANOVA followed by Dunnett's test) and amplitude (**g**, CCI (red) = 16.75 ± 1.102 pA, iβARK (blue) = 13.00 ± 0.5591 pA, mCherry (pink) = 17.80 ± 1.188 pA, $p = 0.0001$, F = 14.22, one-way ANOVA followed by Dunnett's test) in CCI, iβARK and mCherry mice. Cumulative probability plots of mEPSCs inter-event intervals (IEI) (**f**, CCI (red) vs. iβARK (blue), $p < 0.0001$, Kolmogorov-Smirnov test; CCI (red) vs. mCherry (pink), $p = 0.3480$, Kolmogorov-Smirnov Test) and mEPSCs amplitude (**h**, CCI (red) vs. iβARK (blue), $p < 0.0001$, Kolmogorov-Smirnov test; CCI (red) vs. mCherry (pink), $p = 0.1779$, Kolmogorov-Smirnov Test) in CCI, iβARK and mCherry mice. **i** Representative eEPSC traces evoked by pair pulse protocol (PPP) with an interval of 50 ms recorded in the ACC in CCI (red), iβARK (blue), and mCherry (pink) mice. **j** Summary bar graph of paired-pulse ratio (PPR) index ($n = 8$ cells from 4 CCI mice; $n = 7$ cells from 3 iβARK mice; $n = 7$ cells from 3 mCherry mice; CCI (red) = 1.040 ± 0.04728, iβARK (blue) = 1.402 ± 0.1040, mCherry (pink) = 0.9640 ± 0.04784, $p = 0.0007$, F = 10.91, one-way ANOVA followed by Dunnett's test) of eEPSCs in CCI, iβARK and mCherry mice. **k** Extracellular glutamate concentrations in the dialysate in the ACC of CCI, iβARK and mCherry mice ($n = 5$ CCI mice; $n = 6$ iβARK mice; $n = 6$ mCherry mice; CCI (red) = 8.656 ± 0.4981 μM, iβARK (blue) = 5.665 ± 0.3399 μM, mCherry (pink) = 8.117 ± 0.6241 μM, $p = 0.0019$, F = 10.10, one-way ANOVA followed by Dunnett's test). **l** iβARK significantly increased hind paw withdrawal threshold in neuropathic pain mice ($n = 6$ CCI mice; $n = 8$ iβARK mice; $n = 6$ mCherry mice; Day0_CCI (red circle) = 1.567 ± 0.1585 g, Day0_iβARK (blue circle) = 1.367 ± 0.1579 g, Day0_mCherry (pink circle) = 1.494 ± 0.2304 g, Day14_CCI (red circle) = 0.6078 ± 0.1093 g, Day14_iβARK (blue circle) = 1.533 ± 0.2397 g, Day14_mCherry (pink circle) = 0.5956 ± 0.09532 g; Day14_CCI vs. Day14 iβARK, $p = 0.0018$, Day14_CCI vs. Day14 mCherry, $p = 0.9985$, two-way ANOVA (F (2, 17) = 14.14, $p = 0.0002$), Dunnett's post hoc test). Error bars represent the mean ± SEM.

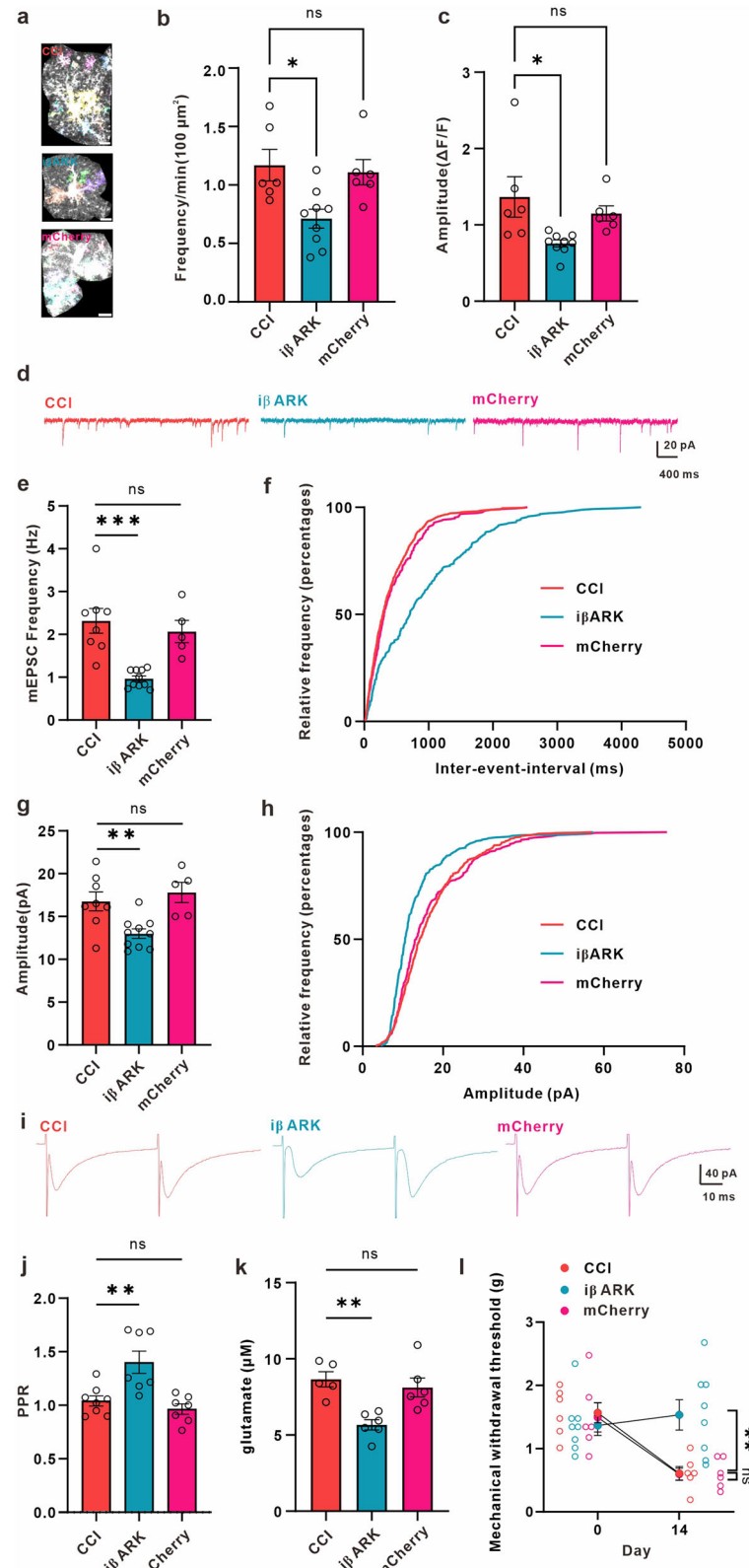

with the concept that the re-emergence of mGluR5 is associated with derived synaptic plasticity and chronic pain, supporting earlier suggestions[31]. Koizumi and colleagues also provided evidence that reactive astrocytes enhance synaptic transmission in chronic pain through the release of Thrombospondin-1, Glypican-4, and Hevin. Whether this

mechanism underlies synaptic plasticity in ACC in our model remains to be investigated.

Overwhelming evidence substantiates that astrocyte calcium plays a crucial role in regulating synaptic plasticity by releasing different glio-transmitters, including glutamate, D-serine, and ATP/adenosine[34,63].

**Fig. 5 | Chemogenetic activation of astrocytes increases synaptic transmission trough a pre-synaptic mechanism and aggravates mechanical allodynia in neuropathic pain mice.**
**a** Representative traces of miniature excitatory postsynaptic currents (mEPSCs) recorded from layer II-III neurons of ACC before (gray) and after (yellow) CNO application in sham mice. (b, d) Summary of mEPSCs frequency (**b**, $n$ = 9 cells from 3 sham mice; baseline (gray) = 1.215 ± 0.1034 Hz, CNO (yellow) = 1.830 ± 0.1595 Hz, $p$ = 0.0273, paired t-test) and amplitude (**d**, baseline (gray) = 10.40 ± 0.8898 pA, CNO (yellow) = 10.30 ± 0.7373 pA, $p$ = 0.8131, paired t-test) before and after CNO application in sham mice. Cumulative probability plots of mEPSCs inter-event intervals (IEI) (**c**, $p$ = 0.0001, Kolmogorov-Smirnov Test) and mEPSCs amplitude (**e**, $p$ = 0.0946, Kolmogorov-Smirnov Test) before (gray) and after (yellow) CNO application in sham mice. **f** Representative eEPSC traces evoked by pair pulse protocol (PPP) with an interval of 50 ms recorded in the ACC before (gray) and after (yellow) CNO application in sham mice. **g** Summary bar graph of paired-pulse ratio (PPR) index ($n$ = 6 cells from 3 sham mice; baseline (gray) = 1.388 ± 0.08709, CNO (yellow) = 1.145 ± 0.07766, $p$ = 0.0425, paired t-test) of eEPSCs before and after CNO application in sham mice. **h** Schematic representation of the experimental design. **i** CNO does not significantly affect hind paw withdrawal threshold in sham-operated mice ($n$ = 5 sham mice; Day0_sham (white circle) = 1.600 ± 0.1211 g, Day14_sham (gray circle) = 1.720 ± 0.3762 g, CNO_sham (blue circle) = 1.533 ± 0.1660 g, $p$ = 0.6466, F = 0.3172, Repeated measures ANOVA summary). **j** CNO significantly decreases hind paw withdrawal threshold in neuropathic pain mice ($n$ = 6 CCI mice; Day0_CCI (white circle) = 1.589 ± 0.1434 g, Day14_CCI (gray circle) = 0.6333 ± 0.1202 g, CNO_CCI (blue circle) = 0.3994 ± 0.09247 g, $p$ < 0.0001, F = 96.43, Repeated measures ANOVA summary). Error bars represent the mean ± SEM.

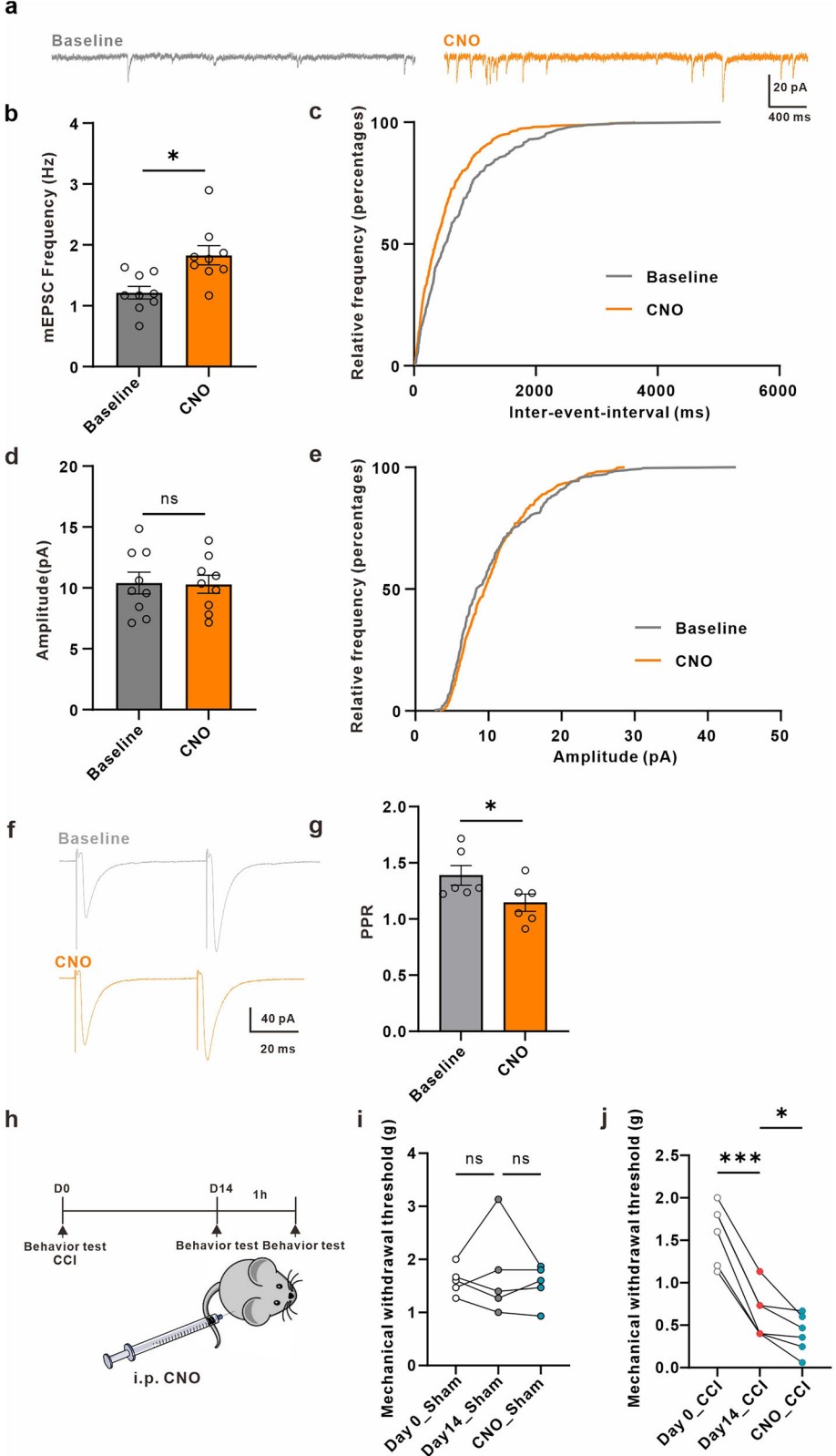

Previous studies have demonstrated that the activation of astrocytic mGluR5 leads to the release of adenosine, which in turn activates adenosine $A_{2A}$ receptors ($A_{2A}$Rs) and enhances excitatory synaptic transmission in the hippocampus[64]. However, other studies indicated that the activation of astrocytic mGluR5 leads to the release of glutamate[65,66]. In mammals and Drosophila, the activation of astrocytic mGluR5 leads to the release of D-serine from astrocytes and further modulates the function of NMDA receptors[67,68]. In the present study, we observed that the amplitude and frequency of mEPSC were increased in the ACC after nerve injury. Additionally, there was an enhancement of PPR, suggesting the involvement of

**Fig. 6 | Down-regulation of mGluR5 in ACC astrocytes decreases astrocytic Ca²⁺ signals in CCI mice. a** Schematic representation of the experimental design. **b** Immunohistochemical images of the astrocytes in the ACC of CCI mice. Upper panel: colocalization of mCherry with the astrocytic-specific marker s100β (green) in the ACC (b1, 90.21 ± 4.904%, $n = 6$ slices from 3 mice); middle and below panel: immunohistochemical staining of astrocyte shRNA (red) and mGluR5 (green) in the ACC. Quantification of mGluR5 fluorescence intensity in shRNA mCherry-expressing astrocytes relative to those expressing scramble-shRNA mCherry (b2, $n = 7$ slices from 3 scramble-shRNA mice, $n = 6$ slices from 3 shRNA mice, scramble-shRNA (pink) = 50.10 ± 1.599 a.u., shRNA (blue) = 30.05 ± 7.04 a.u., $p = 0.0122$, unpaired t-test). **c** Summary plots illustrating that Ca²⁺ signals elicited by CHPG are attenuated by expression of shRNA ($n = 10$ cells from 3 shRNA mice; $n = 8$ cells from 3 scramble-shRNA mice; Baseline_shRNA (blue circle) = 0.003342 ± 0.002943 ΔF/F, CHPG_shRNA (blue circle) = 0.07953 ± 0.02701 ΔF/F, baseline_scramble-shRNA (pink circle) = 0.05103 ± 0.1469 ΔF/F, CHPG_ scramble-shRNA (pink circle) = 0.3477 ± 0.1157 ΔF/F, $p = 0.5003$, two-way ANOVA (F (1, 16) = 12.79, $p = 0.0025$), Fisher's LSD post hoc test). **d** Representative images displaying all AQuA-detected events from a 1-minute ex vivo astrocytic GCaMP7b Ca²⁺ imaging experiment from different groups. Upper: representative image in CCI mice; middle: representative image in shRNA-expressing mice; below: representative imaging in scramble-shRNA-expressing mice; Colors indicate detected events. Scale bar: 10 μm. **e** Graph depicting the frequency of astrocytic Ca²⁺ signals in the ACC of CCI, shRNA and scramble-shRNA mice ($n = 8$ cells from 3 CCI mice; $n = 8$ cells from 3 shRNA mice; $n = 6$ cells from 3 scramble-shRNA mice; CCI (red) = 1.324 ± 0.09899 min/100 μm², shRNA (blue) = 0.5763 ± 0.07242 min/100 μm², scramble-shRNA (pink) = 1.240 ± 0.2585 min/100 μm², $p = 0.0021$, F = 8.690, one-way ANOVA followed by Dunnett's test). **f** Graph showing the amplitude of astrocytic Ca²⁺ signals in the ACC of CCI, shRNA and scramble-shRNA mice (CCI (red) = 1.556 ± 0.2733 ΔF/F, shRNA (blue) = 0.7032 ± 0.09216 ΔF/F, scramble-shRNA (pink) = 1.144 ± 0.06367 ΔF/F, $p = 0.0109$, F = 5.781, one-way ANOVA followed by Dunnett's test). Error bars represent the mean ± SEM.

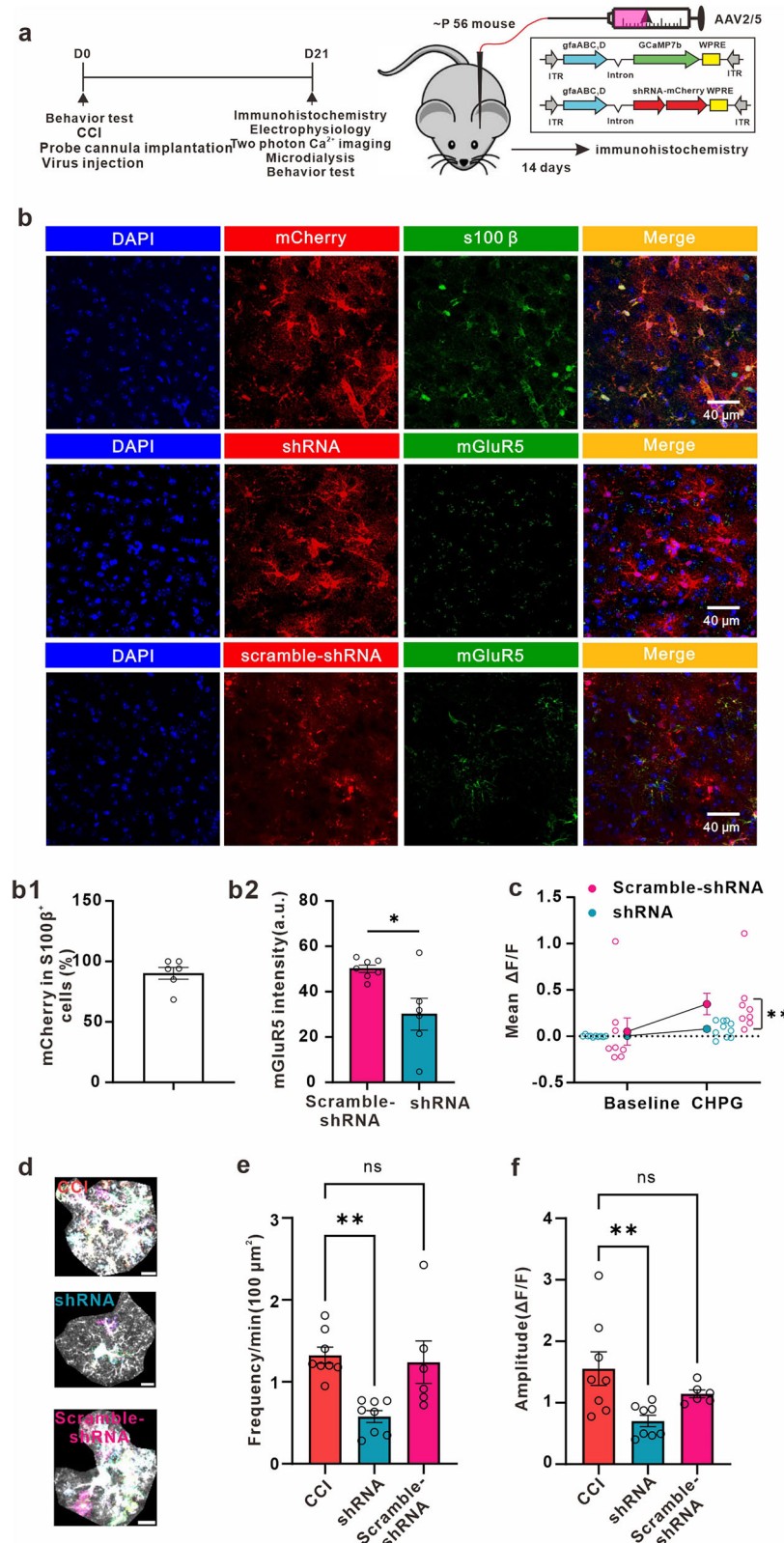

both presynaptic and postsynaptic mechanisms. However, we did not explore whether ATP /adenosine, glutamate, or D-serine is released from astrocytes with Ca²⁺ elevations and whether an autocrine mechanism is involved in the hyperactivity of astrocytes. The precise mechanism underlying the relationship between astrocyte hyperactivity and the enhancement of synaptic transmission is still not fully understood. Therefore, further studies are necessary to investigate the extracellular concentration of gliotransmitters using microdialysis in ACC after peripheral nerve injury.

On the other hand, the specific type of receptor on neurons involved in modulating synaptic transmission remains unclear. Previous studies

**Fig. 7 | Down-regulation of mGluR5 in ACC astrocytes decreases synaptic transmission, extracellular concentration of glutamate, and increases mechanical withdrawal threshold in CCI mice. a** Representative traces of miniature excitatory postsynaptic currents (mEPSCs) recorded from layer II-III neurons of ACC in CCI (red), shRNA (blue) and scramble-shRNA (pink) mice. Summary of mEPSCs frequency (**b**, $n = 6$ cells from 3 CCI mice; $n = 6$ cells from 3 shRNA mice; $n = 7$ cells from 3 scramble-shRNA mice; CCI (red) = 2.072 ± 0.1933 Hz, shRNA (blue) = 1.189 ± 0.1084 Hz, scramble-shRNA (pink) = 2.407 ± 0.3412 Hz, $p = 0.0096$, F = 6.296, one-way ANOVA followed by Dunnett's test) and amplitude (**d**, CCI (red) = 17.98 ± 1.892 pA, shRNA (blue) = 12.24 ± 1.219 pA, scramble-shRNA (pink) = 18.24 ± 1.410 pA, $p = 0.0229$, F = 4.829, one-way ANOVA followed by Dunnett's test) in CCI, shRNA and scramble-shRNA mice. Cumulative probability plots of mEPSCs inter-event intervals (IEI) (**c**, CCI (red) vs. shRNA (blue), $p < 0.0001$, Kolmogorov-Smirnov test; CCI (red) vs. scramble-shRNA (pink), $p = 0.4771$, Kolmogorov-Smirnov Test) and mEPSCs amplitude (**e**, CCI (red) vs. shRNA (blue), $p < 0.0001$, Kolmogorov-Smirnov test; CCI (red) vs. scramble-shRNA (pink), $p = 0.4006$, Kolmogorov-Smirnov Test) in CCI, shRNA and scramble-shRNA mice. **f** Representative eEPSC traces evoked by pair pulse protocol (PPP) with an interval of 50 ms recorded in the ACC in CCI (red), shRNA (blue) and scramble-shRNA (pink) mice. **g** Summary bar graph of paired-pulse ratio (PPR) index ($n = 7$ cells from 7 CCI mice; $n = 7$ cells from 4 shRNA mice; $n = 9$ cells from 3 scramble-shRNA mice; CCI (red) = 0.9674 ± 0.04246, shRNA (blue) = 1.653 ± 0.3287, scramble-shRNA (pink) = 1.064 ± 0.1005, $p = 0.0427$, F = 3.709, one-way ANOVA followed by Dunnett's test) of eEPSCs in CCI, shRNA and scramble-shRNA mice. **h** Extracellular glutamate concentrations in the dialysate in the ACC of CCI, shRNA and scramble-shRNA mice ($n = 5$ CCI mice; $n = 7$ shRNA mice; $n = 6$ scramble-shRNA mice; CCI (red) = 9.904 ± 0.9235 μM, shRNA (blue) = 5.033 ± 0.7157 μM, scramble-shRNA (pink) = 8.943 ± 0.8825 μM, $p = 0.0017$, F = 10.09, one-way ANOVA followed by Dunnett's test). **i** Knockdown mGluR5 in ACC astrocytes significantly increased hind paw withdrawal threshold in neuropathic pain mice ($n = 6$ CCI mice; $n = 7$ shRNA mice; $n = 11$ scramble-shRNA mice; Day0_CCI (red circle) = 1.322 ± 0.07585 g, Day0_shRNA (blue circle) = 1.381 ± 0.1906 g, Day0_scramble-shRNA (pink circle) = 1.145 ± 0.1108 g, Day21_CCI (red circle) = 0.5972 ± 0.1801 g, Day21_shRNA (blue circle) = 1.522 ± 0.2658 g, Day21_scramble-shRNA (pink circle) = 0.4139 ± 0.08482 g; Day21_CCI vs. Day21_shRNA, $p = 0.0011$, Day21_CCI vs. Day21_scramble-shRNA, $p = 0.5003$, two-way ANOVA $(F(2, 21) = 4.459$, $p = 0.0243)$, Tukey's post hoc test). Error bars represent the mean ± SEM.

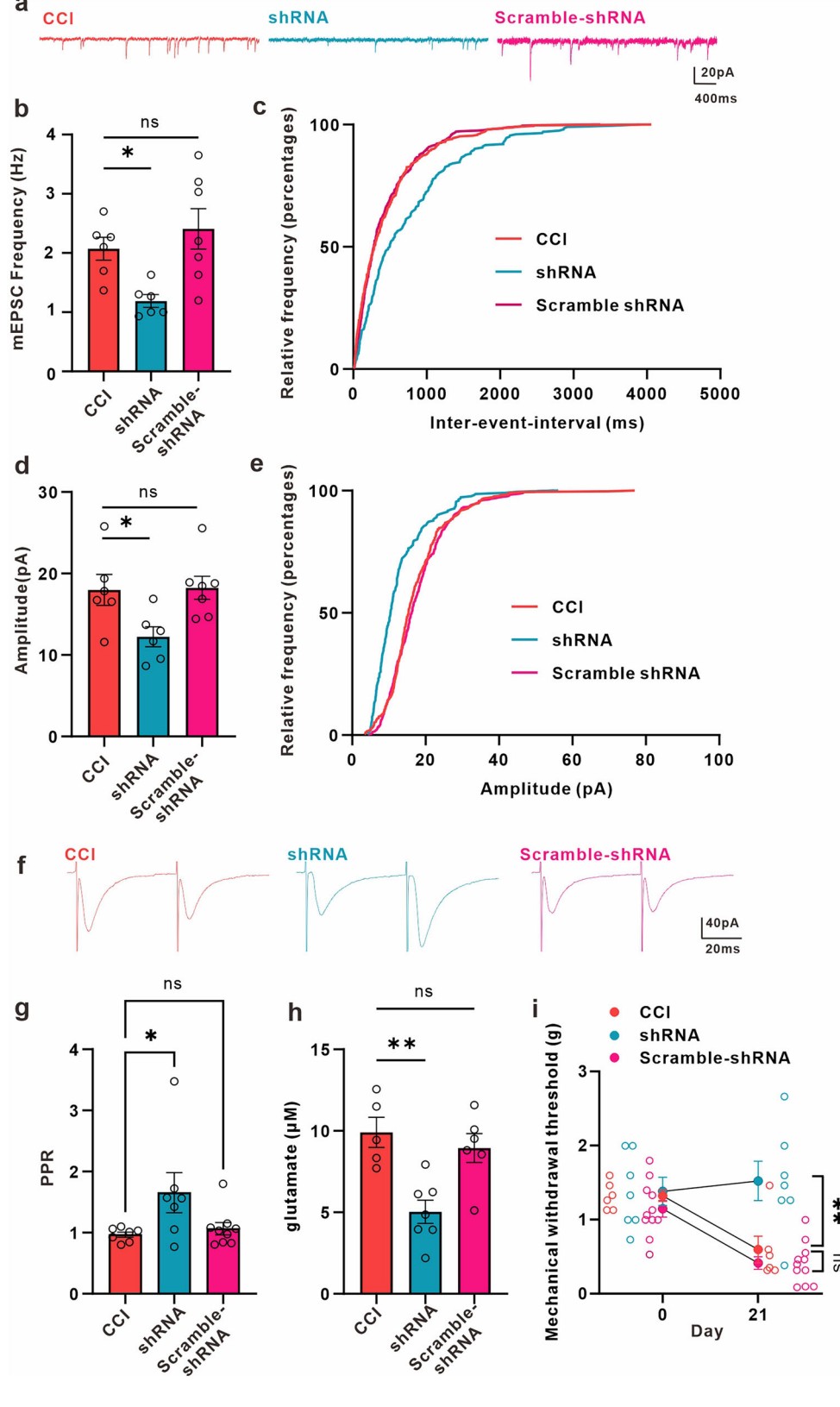

indicated that presynaptic GluN2C/2D-containing NMDA receptors and postsynaptic GluN2A/2B-containing NMDA receptors may be involved in the modulation of presynaptic and postsynaptic activities in the ACC in chronic pain animal models, respectively[69]. It has also been suggested that GluK1-containing kainate receptors are necessary for its induction and the expression of presynaptic LTP in the ACC[1,49]. These results indicate that gliotransmitter glutamate or D-serine could act as a modulator to regulate synaptic transmission in the ACC in a chronic pain model. Further studies are required to validate which receptor on neuron could bind glio-transmission and modulate synaptic transmission

Mounting evidence indicates that LTP and long-term depression (LTD) in the dorsal horn of the spinal cord and cortical regions, such as ACC, play a causal role in chronic pain[1,70,71]. There is ample evidence from animal models indicating that LTP at C-fiber synapses in the spinal cord can contribute to hyperalgesia[72]. In the present study, we demonstrated that activation of astrocytes in the ACC using chemogenetics does not have an impact on the responses to painful stimuli in normal, non-injured mice. However, upon activation of ACC astrocytes in mice with nerve injuries, we observed an exacerbation of mechanical allodynia. These observations can be explained by the occurrence of LTP at C-fiber synapses in the spinal cord, which cannot be induced without nerve injury. Thus, in comparison to mice with nerve injuries, the transmission of sensory information related to pain from peripheral tissues through the spinal cord to the ACC is not as readily facilitated in non-injured mice. In line with this, our findings suggest that cortical astrocyte activation alone does not appear sufficient to induce pain-related responses under normal conditions. However, in the context of neuropathic pain, where spinal LTP and other pain-sensitizing mechanisms are already engaged, astrocyte activation in the ACC may contribute to the amplification of pain signals. This indicates that astrocytes in the cortex may only adopt a pain-facilitatory role in the presence of underlying neuropathic changes, rather than being intrinsically activated to promote pain.

Nevertheless, there are still many intriguing questions that have yet to be answered. Although we and others show that neuropathic pain increases the mGluR5 levels in cortex astrocytes, the molecular mechanisms controlling the reemergence of mGluR5 require further research.

Our findings reveal the role of mGluR5 deletion in rescuing mechanical allodynia. This is achieved by protecting ACC pyramidal neurons from the excess glutamate transmission that arises under chronic pain conditions due to the hyperactivity of astrocytes. These results establish a causal relationship between aberrant astrocytic calcium signaling in the ACC and neuropathic pain in male mice. Furthermore, mGluR5 may provide new insights into neuropathic pain by targeting neuron-glia interactions.

## Limitations of study

This study provides valuable insights into the role of mGluR5-mediated astrocytic hyperactivity in the ACC in neuropathic pain, but several limitations must be acknowledged:

The study exclusively used male mice to avoid variability due to estrous cycle influences on pain sensitivity. However, this approach does not reflect the clinical prevalence of chronic pain, which affects more women than men and is often more severe in women[73,74]. There is increasing evidence of sex differences in mechanisms of neuropathic pain, including glia-mediated pathways[75–77]. This limitation reduces the generalizability of our findings and highlights the need for future research to include female subjects to explore potential sex-specific mechanisms.

An important limitation of this study is the presence of outliers in the mechanical pain threshold data, particularly in Fig. 5i, where reported thresholds exceeded physiologically plausible values (over 2 g). These outliers may skew the statistical analysis, leading to artificial significance or misinterpretation of the data. Another important limitation of this study is that some analyses were conducted without experimenter blinding. This raises concerns about potential bias in data interpretation. To address this, future studies should implement rigorous blinding protocols to enhance the reliability and robustness of the findings.

## Methods

### Animals

Our study was conducted in accordance with the Guide for the Care and Use of Laboratory Animals and received approval from the ethics committee of Hangzhou City University (registration number: 22061). We have complied with all relevant ethical regulations for animal use. Male C57BL/6 mice (~2 months old) were purchased from Hangzhou Ziyuan Laboratory Animal Corporation and were housed in groups of three to four per cage, with softened paper for enrichment in cages (35 cm in length, 13 cm in width, and 13 cm in height). The mice were maintained on a 12-hour light/dark cycle and given unrestricted access to food and water. We randomly tagged each mouse and then assigned them to groups at random. Only male mice were used in this study to avoid potential variability in pain sensitivity associated with the estrous cycle in females, which could influence the results of pain threshold measurements.

### Surgery

The chronic constriction injury model was adapted from a previously established protocol with slight modifications[78]. The 2-month-old male mice were anesthetized with isoflurane (1-2% concentration, with an airflow rate of 0.4 L/min) and positioned laterally on a warming blanket. The hair on the left hind leg was removed using a clipper. After disinfecting the skin above the left hind limb with iodophor, a longitudinal incision of approximately 1.5 cm in length was made. The muscle was bluntly dissected to expose the sciatic nerve. Three loosely constrictive ligatures (6-0 chromic gut suture) were gently tied around the nerve, with a spacing of 1–1.5 mm between each ligature. The appropriate tension for ligation was determined by observing slight tremors in the mice. In the sham group, the sciatic nerve was exposed using the same method, but without ligation. After ligation, the muscles and skin were sutured layer by layer, and the sutured skin was disinfected again with iodophor.

### Mechanical allodynia test

For the measurement of mechanical pain threshold, the mice were placed in an organic glass chamber (11.5 cm long, 11.5 cm wide, 13 cm high) with a metal grid at the bottom. They were allowed to acclimate for at least 60 min to minimize stress and ensure accurate measurement of baseline responses. After the mice adapted to the environment, Von Frey filaments were used to stimulate the plantar surface of the left hind paw. The stimulation intensity was applied in ascending order, starting from low to high. Each filament was bent to a 90-degree angle and applied for 4–5 seconds.

A positive response was recorded if the mouse exhibited immediate paw withdrawal or avoidance reactions during or following the stimulation. If no such reaction was observed, it was recorded as a negative response. The paw withdrawal threshold (PWT) was determined as the lowest filament intensity that elicited 3 or more positive responses out of 5 tests. If fewer than 3 positive responses were recorded, the next filament with a higher intensity was used, and the process was repeated until the mechanical pain threshold was identified. Each test was repeated 3 times, with a minimum of 5 min between repetitions to prevent sensitization. The final PWT was calculated as the average of the 3 tests. To eliminate bias, the observer conducting the test was blinded to the experimental conditions to ensure an objective assessment of paw withdrawal responses.

### Stereotactic virus injection

For stereotactic virus injection, mice were anesthetized with isoflurane (1–2% concentration, with an airflow rate of 0.4 L/min), and their heads were then immobilized using a stereotaxic frame (RWD, 68930). A warming blanket was placed underneath to maintain their body temperature at physiological levels. After shaving off their hair and sterilizing the skin with iodophor, an incision was made using surgical scissors to expose the skull. The surface connective tissue on the skull was removed using $H_2O_2$. The skull was allowed to dry. Using a dental drill, a hole was drilled in the skull at a position 0.7 mm

https://doi.org/10.1038/s42003-025-07733-5 **Article**

anterior to the bregma and 0.3 mm lateral to the midline (right). The dura mater was carefully removed using fine forceps.

The viral constructs used were as follows: AAV2/5-GfaABC1D-jGCaMP7b-WPRE with a titer of $3 \times 10^{12}$ gc/ml for calcium imaging, AAV2/5-GfaABC1D-iβARK-2A-mCherry with a titer of $5.94 \times 10^{12}$ gc/ml for blocking Gq-protein coupled receptor-mediated $Ca^{2+}$ signaling, AAV2/5-GfaABC1D-mCherry with a titer of $5.09 \times 10^{12}$ gc/ml for mCherry expression, and AAV2/5-GfaABC1D-hM3D(Gq)-mCherry with a titer of $5.31 \times 10^{12}$ gc/ml for chemogenetic manipulation. In order to knockdown mGluR5 expression in ACC astrocytes, we utilized AAV2/5-GfaABC1D-mGluR5 shRNA ($6.4 \times 10^{12}$ gc/ml) containing the shRNA sequence (5'-GCAGCTTGCATCGCCTATATC-3'). When injecting a single virus, 200 nL of the virus was slowly infused into the ACC of the mouse at a rate of 1 nL/s using a microinjection pump (injection site at a depth of 1.3 mm from the brain). When performing co-injection of two viruses, 100 nL of each virus was mixed thoroughly and then injected into the ACC at a rate of 1 nL/s. After the virus injection, the micropipette needle was left in place in the target brain area for 15 min to allow for proper viral diffusion. The mice were then allowed to recover in individual cages for further study. All viruses were generated by Brainvta, Braincase, and Sunbio Medical Biotechnology (Key Resources Table).

## Immunohistochemistry

For immunohistochemistry mice were first transcardially perfused with PBS, followed by perfusion with 4% paraformaldehyde (PFA) to fix the brain tissue. The whole brain was then extracted and placed in 4% PFA solution for an additional 24 h of fixation. Subsequently, the brain tissue containing the ACC was sectioned into 50 μm thick slices using a vibratome (Leica VT1200 S). The brain slices were washed three times for 5 min each in 0.1 M PBS (pH 7.4). The slices were then blocked with a mixture of 4% normal goat serum (NGS) and 1% Triton-X in PBS at room temperature for 60 min. After blocking, the slices were washed three times with PBS and incubated overnight at 4 °C with specific primary antibodies (S100β antibody at 1:1000 dilution, mGluR5 antibody at 1:500 dilution, NeuN antibody at 1:500 dilution, Iba1 antibody at 1:500 dilution, NG2 antibody at 1:500 dilution). The details of the antibodies can be found in the Key Resources Table. Following primary antibody incubation, the slices were washed three times with PBS and incubated with fluorescence-labeled secondary antibodies at room temperature for 2 h. Finally, the slices were washed with PBS and mounted on glass slides. Images were examined using a confocal laser scanning microscope (Olympus, VT1000 and Zeiss LSM 980) and analyzed using ImageJ (NIH, RRID: SCR_003070). For the analysis of mGluR5 fluorescence intensity, the mean gray value was used as the measurement

| Key Resources Table | | | | |
|---|---|---|---|---|
| **Reagent type (species) or resource** | **Designation** | **Source or reference** | **Identifiers** | **Additional information** |
| Strain | C57BL/6 male mice (mus musculus) | Hangzhou Ziyuan Laboratory Animal Corporation (RRID:IMSR_JAX:000664) | N/A | |
| Genetic reagent (virus) | AAV2/5 GfaABC1D jGCaMP7b | Brain Case (http://www.braincase.cn/) | N/A | $3 \times 10^{12}$ gc/ml |
| Genetic reagent (virus) | AAV2/5 GfaABC1D- iβARK-2A-mCherry | Brainvta (https://www.brainvta.tech/) | N/A | $5.94 \times 10^{12}$ gc/ml |
| Genetic reagent (virus) | AAV2/5 GfaABC1D- mCherry | Brainvta (https://www.brainvta.tech/) | N/A | $5.09 \times 10^{12}$ gc/ml |
| Genetic reagent (virus) | AAV2/5 GfaABC1D- shRNA-mCherry | Sunbio Medical Biotechnology (http://www.sbo-bio.com.cn/) | 5'-GCAGCTTGCATCGCCTATATC-3' | $6.4 \times 10^{12}$ gc/ml |
| Genetic reagent (virus) | AAV2/5 GfaABC1D- scramble-shRNA-mCherry | Sunbio Medical Biotechnology (http://www.sbo-bio.com.cn/) | 5'-GTTCTCCGAACG TGTCAC GTA-3' | $5.1 \times 10^{12}$ gc/ml |
| Chemical compound, drug | picrotoxin | MedChemExpress (https://www.medchemexpress.com/) | N/A | 100 μM |
| Chemical compound, drug | tetrodotoxin (TTX) | (R&D System)https://www.rndsystems.com/cn | N/A | 1 μM |
| antibody | mouse monoclonal anti-s100β | Synaptic systems Cat #287111 | RRID:AB_2814886 | 1/1000 |
| antibody | mouse monoclonal anti-NeuN | Millipore Cat# MAB377 | RRID: AB_2298772 | 1/500 |
| antibody | Rabbit monoclonal anti-mGluR5 | CST Cat# 55920 | RRID:AB_2734718 | 1/500 |
| antibody | Rabbit monoclonal anti-Iba1 | FUJIFILM Wako Pure Chemical Corporation Cat# 019-19741 | RRID:AB_839504 | 1/500 |
| antibody | Rabbit monoclonal anti-NG2 | Millipore Cat# AB5320 | RRID:AB_91789 | 1/250 |
| software, algorithm | AQuA (Astrocyte Quantification and Analysis) | N/A | https://github.com/yu-lab-vt/AQuA?tab=readme-ov-file | |
| software, algorithm | Coreldraw | Alludo | https://www.coreldraw.com | |
| software, algorithm | Prism | Graphpad | https://www.graphpad.com/scientific-software/prism/ | |
| software, algorithm | Matlab | MathWorks | https://se.mathworks.com/products/matlab.htm | |
| software, algorithm | Fiji/ImageJ | NIH | https://imagej.nih.gov/ij/ | |

**Article**

parameter. The mean gray value was calculated using the formula: Mean Gray Value = Integrated Density/Area.

## Electrophysiology

For electrophysiology, mice were anesthetized using isoflurane, euthanized via cervical dislocation followed by decapitation, and their brains were quickly extracted and immersed in an ice-cold solution with the following composition (in mM): 235 Sucrose, 1.25 NaH$_2$PO$_4$, 2.5 KCl, 0.5 CaCl$_2$, 7 MgCl$_2$, 10 Glucose, 26 NaHCO$_3$, and 5 Pyruvate (pH 7.3, 310 mOsm, saturated with a mixture of 95% O$_2$ and 5% CO$_2$). Coronal slices of the ACC, 300 μm in thickness, were prepared using a vibrating slicer (Leica, V T1200). The slices were then incubated for approximately 30 min in artificial cerebrospinal fluid (ACSF) with the following concentrations (in mM): 26 NaHCO$_3$, 2.5 KCl, 126 NaCl, 10 D-glucose, 1 sodium pyruvate, 1.25 NaH$_2$PO$_4$, 2 CaCl$_2$, and 1 MgCl$_2$. The ACSF had a pH of 7.4 and an osmolality of 310 mOsm. It was saturated with a mixture of 95% O$_2$ and 5% CO$_2$.

The brain slice was moved to a recording chamber and fixed with a grid. Oxygen-saturated ACSF was perfused into the chamber at a rate of 3.0 ml/min, and the cells on the brain slice were imaged on a monitor screen using a differential interference contrast imaging system and an infrared conversion camera. The recording electrode was pulled from borosilicate glass pipettes (outer diameter 1.5 mm, inner diameter 0.86 mm) using a horizontal microelectrode puller (Sutter, P-97), with a tip impedance of approximately 3–5 MΩ. The internal solution for neuronal recordings in voltage-clamp mode contained (in mM): 125 CeMeSO$_3$, 10 HEPES, 1 MgCl$_2$, 1 CaCl$_2$, 8 TEA-Cl, 5 4-AP, 0.4 Na$^+$GTP, 4 ATP-Na$_2$, 10 EGTA, pH adjusted to 7.35.

The paired-pulse ratio (PPR) was recorded in Layer II-III pyramidal neurons of the ACC using whole-cell mode at -70 mV. Electrical stimuli were delivered every 20 s via a bipolar tungsten electrode in the fifth layer of the ACC, with an interstimulus interval of 50 ms. The PPR, which is calculated as the ratio of the second pulse-evoked excitatory postsynaptic current (EPSC$_2$) to the first pulse-evoked EPSC (EPSC$_1$), reflects presynaptic glutamate release. A decrease in PPR indicates an increase in presynaptic glutamate release, while an increase in PPR indicates a decrease in presynaptic glutamate release.

Miniature excitatory postsynaptic currents (mEPSCs) were recorded in whole-cell mode at -70 mV in layer II-III pyramidal neurons of ACC brain slices. Picrotoxin (100 μM) and tetrodotoxin (TTX, 1 μM) were used to block network activity and inhibitory synaptic transmission. The details of the reagents can be found in the Key Resources Table.

Axopatch 700B amplifiers (Molecular Devices) were used for patch-clamp recordings. The recorded data was filtered at 6 kHz and sampled at a rate of 20 kHz. The pClampfit 10.6 program (Molecular Devices) was utilized for offline data processing. To monitor series resistances and membrane resistances, negative pulses (-10 mV) were applied. The aim was to determine if the series resistances remained stable throughout the experiment, with a maximum allowable variation of less than 20%. All experiments were conducted at a temperature of 32°C.

## Microdialysis

The mice were anesthetized with isoflurane and positioned in a stereotaxic frame. A guide cannula for the microdialysis probe was implanted into the ACC region at the coordinates AP, +0.7 mm; ML, −0.3 mm; DV, 1.7 mm relative to bregma. The cannula was fixed to the skull using dental cement and screws. After surgery, mice were allowed to recover for 14 days in individual cages. On the day of the experiment, the mice were gently restrained, and the guide cannula was exposed. A microdialysis probe (AZ-4-02, EICOM Corp.) was inserted through the guide cannula and gradually lowered into the ACC region 2 h prior to the initiation of microdialysis. The probe was connected to a microinjection pump through a microdialysis tubing system. aCSF was perfused through the probe at a constant flow rate of 1.5 μl/min. Once a stable baseline (30 min) was established, dialysate samples were collected for 40 min. Dialysate samples were collected in

microcentrifuge tubes and promptly stored at -80°C for further analysis. The concentration of extracellular glutamate in the samples was quantified with a glutamate assay kit (ab83389, Abcam) according to the protocol. After experiments, the positions of the cannulae were verified. Mice with incorrect positioning of cannulae were excluded.

## Two-photon Ca$^{2+}$ imaging

Ca$^{2+}$ signals in ACC astrocytes were observed using a two-photon system (FVMPE-RS; Olympus) equipped with a 25x water immersion objective lens (NA = 1.05). Brain slices were perfused continuously with oxygenated ACSF containing TTX (1 μM) and picrotoxin (100 μM) at a rate of 3.0 ml/min. Excitation of GCaMP7b and mCherry was achieved at 920 nm and 1100 nm, respectively. Astrocytes located at least 40 μm away from the slice surface in the ACC were selected for imaging. Imaging for spontaneous and evoked Ca$^{2+}$ signals in astrocytes was performed at a rate of 1 frame per 1000 ms. Spontaneous Ca$^{2+}$ signals were acquired continuously for 5–10 mins. Activation of hM3D(Gq)-mediated Ca$^{2+}$ signals and mGluR5 were achieved by bath application of CNO (5 μM) and (RS)-2-chloro-5-hydroxyphenylglycine (CHPG, 500 μM), respectively. In some experiments, image stacks comprising optical sections spaced 1 μm apart were acquired (10–20 sections total) to aid in identifying co-expression of GCaMP7b and hM3Dq (mCherry) in astrocytes. To eliminate any x-y drift, movies were registered using the StackReg plugin in ImageJ software. The analysis of evoked Ca$^{2+}$ signals followed previously described methods. Spontaneous Ca$^{2+}$ signals were analyzed with AQuA (Astrocyte Quantification and Analysis)[79]. For evoked Ca$^{2+}$ signals, selected regions of interest (ROIs) were analyzed using the Time Series Analyzer V3 plugin in Fiji/ImageJ. ΔF/F values for GCaMP7b fluorescence were calculated as ΔF/F = (F-F0)/F0, where F represents fluorescence intensity and F0 represents baseline fluorescence intensity. The mean ΔF/F values, as well as the amplitude and frequency of the calcium signals, were analyzed using Clampfit 10.6 software and AQuA.

## Statistics and reproducibility

All data processing, figure generation, layout, and statistical analysis were performed using Clampfit 10.6, GraphPad Prism, MATLAB and Coreldraw. The sample size for this study was determined based on previous studies from our laboratories. To assess the normality of the data, the Shapiro-Wilk test was used. The results of the test were not statistically significant ($p > 0.05$), the data were assumed to follow a normal distribution. To statistically analyze cumulative frequency distributions, the Kolmogorov-Smirnov test was employed. When comparing two groups, Student's $t$-test (paired or unpaired) was used. When comparing three groups, One-way repeated measures (RM) ANOVA followed by Dunnett's post-hoc test or two-way ANOVA was used.

All values were reported as mean ± standard error of the mean (SEM). Statistical significance was determined when the $p$-value was less than 0.05 (*$p < 0.05$, **$p < 0.01$, ***$p < 0.001$). The sample sizes for each analysis were indicated in the figure legends.

## Reporting summary

Further information on research design is available in the Nature Portfolio Reporting Summary linked to this article.

## Data availability

Data are available from the corresponding author on reasonable request. Numerical source data for all graphs in the manuscript can be found in supplementary data 1 file.

## Code availability

Code is available from the corresponding author on reasonable request.

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

## Acknowledgements

We acknowledge BioRender (https://biorender.com/) for providing the tools to create the graphical abstract in this manuscript. This work was supported by grants from Zhejiang Provincial Natural Science Foundation of China (LQ23C090001), key R&D Program Project of Zhejiang (2022C03034), Scientific Research Foundation of Hangzhou City University (J-202325, F-202405, X202304), National College Students' Innovative Entrepreneurial Training Plan Program (202313021034, 202313021026), The Zhejiang Provincial Medical and Health Science and Technology Plan (2025KY1576), Xinhua Hospital, Shanghai Jiaotong University School of Medicine (2021XHYYJJ08).

## Author contributions

W.S. and L. Zeng designed the study; W.S., F.C., Y.T., L. Zeng and Y.Z. analyzed data wrote the manuscript; W.S., F.C., Y.T., Y.Z., L. Zhu and L.N. carriedoutmost experiments; L.X. and L.W. helped with analysis; W.Z. helped with viral injection; Y.C. helped with electrophysiological experiments; W.Z., Y.C., J.L. and H.H. helped with immunohistochemistry. All authors discussed the results and comments on the manuscript.

## Competing interests

The authors declare no competing interests.
