## [Transparent Peer Review file · Communications Biology]

mGluR5-mediated astrocytes hyperactivity in the anterior cingulate cortex contributes to neuropathic pain in male mice

Corresponding Author: Dr Weida Shen

Version 0:

Reviewer comments:

Reviewer #1

(Remarks to the Author)

The manuscript by Shen et al. aims to characterize the effects of chronic neuropathic pain on astrocytes in the anterior cingulate cortex of mice. The authors make the following claims: astrocytes are hyperactive in the context of chronic pain; chronic pain increases extracellular glutamate levels and synaptic transmission in the ACC; chronic pain leads to a reemergence of mGluR5 on astrocytes; astrocyte mGluR5 signaling is directly responsible for the hyperactivity of astrocytes and synaptic transmission as well as for the pain sensation.

Overall, the quality of the research is good and the results will be of interest to researchers from both the glia field and the pain field. However, there are some specific comments that need to be addressed to strengthen the conclusions of the paper.

Major points:

1. The claim is that astrocytes are hyperactive in the CCI condition, however, to determine whether this effect is independent of neuronal activity – you would need to measure the calcium activity in the presence of TTX and picrotoxin as you do later in the article.
2. For the microdialysis experiment, in order to say that the glutamate is released from astrocytes, at a minimum it would need to be done in the presence of TTX. If possible, measurements of glutamate using iGluSnfr would yield more specific results.
3. The mGluR5 reemergence claim needs more support – the antibodies for mGluR5 are notoriously unspecific and the figure on the company website looks very different from the figure in this manuscript. The frequency of events between Figures 1D and 3C are at baseline different, with the MPEP results closer to the ones in 1D. I think a better experiment would be to add CHPG or DHPG to the slice to see the response to mGluR5 agonist in the sham and CCI condition (in the presence of TTX and picrotoxin) and also to add agonist + antagonist (CHPG + MPEP + TTX + PTX) to ensure specificity.
4. The iBARK experiment in Figure 5 needs data from a sham animal to show that iBARK is acting on mGluR5 in the CCI condition.
5. It is unclear to me whether figure 7 and figure 8 are done only in the CCI condition or also in sham. The reason this is important, is that the responses to CNO seem abnormally long in duration as compared to the ones shown in the literature (e.g. Durkee et al., 2019). I think this should be further investigated and addressed.
6. As shRNAs have sometimes off target effects, I would want clearer confirmation that mGluR5 is knocked out specifically in astrocytes – this can be a slice experiment with showing no responses to mGluR5 agonist or instead using an astrocyte specific knock out mouse.

Minor points:

1. Please provide a rationale in the methods section for why only male mice were used in this study.
2. Please provide more detail in the methods section for how paw withdrawal was assessed. Was the observer blind to the conditions of sham vs. CCI? This might be an important point.
3. Some typos and grammatical errors need to be fixed (e.g. line 362)
4. In figure 4b top row, the red and green channel are switched in the merged image.
5. Figure 5a the colors of gcamp and mcherry are switched.
6. Figure 5b what are the values of the colors in the kymograph? And why is there a color bar for the bottom graph?
7. In 5c and 5f it might be useful to use a median instead of the mean. Getting negative values is a little odd.

8. Figure 5 legend is incorrect.
9. Figure 6L and 10I – the statistics used don't make sense. It should be done with a mixed ANOVA for time and manipulation.
10. Figure 7b – I don't see a difference between the calcium before and after the CNO treatment (seems to be a mistake).

Reviewer #2

(Remarks to the Author)

The manuscript, mGluR5-mediated astrocytes hyperactivity in the anterior cingulate cortex contributes to neuropathic pain, investigates the role of astrocyte mGluR5 mediated calcium in neuropathic pain in male mice, using the chronic constriction injury model. Although the findings of the study are potentially interesting, many controls are missing that prevent the strong conclusions drawn. Below are my comments.

- A clear rationale and justification must be provided for using exclusively male mice to investigate a condition where over 65% of the people suffering are women. Not only do more women suffer from chronic pain than men, they report greater severity of symptoms. Moreover, and very pertinent to the current study, mounting evidence indicates sex differences in mechanisms promoting neuropathic pain (Tansley et al., 2022, NatCommunications; Mapplebeck et al., 2016, Pain); sex differences in ACC-mediated chronic pain states have been reported in rodents (Jarrin et al., 2020, Frontiers in Beh Neur); astrocytes appear to be sexually dimorphic (Chowen et al., 1995, Neuroscience; Mong et al., 2022, DevBrainRes); and mGluR mediated astrocyte-neuronal interactions appear to be sex-dependent (Meadows et al., 2024, Cell Reports), particularly mGluR5 has been observed to exert sex-dependent effects on brain function (Fagan et al., 2020, PNAS; Abd-Elrahman et al., 2020, Sci. Signaling). The lack of female mice in this manuscript is a major weakness that reduces the impact of the data.
- Related to above, every statement where the manuscript states findings (i.e. "we observed that chronic pain is associated with an increase in both the frequency and amplitude of mEPSCs in the ACC", MUST be accompanied with the statement 'in male mice'.
- Every statement where the article states "mice" related to the experiments in the manuscript, MUST add 'male' to reflect that this only applies to male mice.
- The title must be changed to say 'in male mice'
- The experiments were not performed with experimenters blind to treatment, which is well known to potentially impact the data, and thus takes away some of the strength of the findings.
- A more thorough description of housing conditions should be provided: what enrichment was available? Size of the cages?
- How much isoflurane was used for all surgeries?
- For immunohistochemistry, was there no saline flush prior to PFA perfusion?
- For the electrophysiology experiments, the manuscript should indicate how the mice were killed prior to removing the brains.
- The timeline for ALL AAV surgeries relative to subsequent experiments needs to be provided.
- How long were mice allowed to recover post cannula surgeries? And how long after was CCI administered?
- Statistical analyses: section states that paired t-tests were used, then states paired or unpaired, or ANOVAs; it seems the first statement should be removed. PRISM (spelled PRIZM in manuscript) is stated in the table provided, but not in the paragraph explaining statistical tests, this should be clarified.
- Behavioral data providing mechanical thresholds for the mice that are used for the data in Figures 1 and 2 should be provided.
- The results for Figure 1 state that GCaMP7b was "expressed specifically in astrocytes" stating that no co-staining was observed in NeuN expressing cells. This indicates that neurons were not expressing this viral construct, however various glial cell types express GFAP, and thus stating that only astrocytes express it is an overstatement considering the data presented. The data simply support that neurons are not expressing it.
- There should be a reference for the statement "Previous research has indicated that chronic pain induces presynaptic long-term potentiation (pre-LTP) via GluK1-353 containing kainate receptors."
- The findings of Figure 3a are not very convincing. The results state that mGluR5 was significantly expressed after 14 days of CCI, yet no values or statistical measures are provided. Furthermore, half of 1 cell/astrocyte expressing mGluR5 is shown in the ACC of mice with CCI, compared to the 0 in the sham group. Presumably, this is a representative image, suggesting that it's barely expressed. How many mice were used for this analysis? How many slices per mouse? Where the images

taken of all layers, or just some layers in the ACC?

- Evidence for selective expression of iBARK and GCaMP7b on astrocytes vs. other glial cells should be provided.
- The conclusions drawn from the MPEP experiments are not fully supported by the data. There is insufficient controls provided to determine if indeed MPEP is affecting astrocytic calcium levels directly. For instance, it could also be due to presynaptic glutamatergic release, as mGluR5 activation can enhance neuronal firing (Li et al., 2014, JNeurosci). Data indicating that MPEP is not affecting neurons should be provided. If neuronal firing was controlled for, this should be clarified.
- The language used in the results section pertaining to data in Figure 6 should be softened. Although glutamate concentrations are reduced, it cannot be stated that it "leads to alleviation of allodynia"; similarly with mCherry controls that show larger glutamate concentrations, it cannot be said that those concentrations cause increased allodynia. Cause and effect experiments would be needed.
- Data for Figure 8I and J are odd: 8i) gram forces greater than 2 grams would lift the paw of the mice up, as mice are simply too small and light for such VonFrey forces; 8j) the interpretation provided is that CNO significantly enhances allodynia in CCI mice. However, the "allodynia" displayed by 14d CCI mice is rather absent. Indeed, earlier, in Figure 6L (also CNO treatment), it was shown that CCI caused thresholds of 0.6078 and 0.5956 g in CCI mice. In figure 8J, at least 2 mice are way above that, with CNO treatment resulting in over half of the mice showing thresholds of above 0.5g, and only 2 lower, with only one providing what could be considered "enhanced allodynia". Moreover, the non-allodynic, large thresholds shown for Day14_CCI skews the experiment towards concluding that CNO decreases thresholds. Notably, the n value provided states 6, but only CNO_CCI shows 6 data points, Day14_CCI shows merely 3; this should be clarified.
- Data of the effects of CNO on CCI mice expressing only mCherry needs to be provided, to conclude that the CNO effects are mediated via DREADDs
- The data presented does not support the conclusion that "The group receiving mGluR5-shRNA delivery showed significantly reduced levels of mGluR5 expression in astrocytes". It shows that mCherry (presumably in the same vectors expressing shRNA) co-localizes with an astrocyte marker. It also shows that shRNA decreases mGluR5. No data is provided showing this decrease is in astrocytes.
- Statistical measures of mGluR5 knockdown should be provided.
- Why is the data in Figure 10 of CCI mice at day 21 when all other data is at day 14? Are the electrophysiological recordings in Figure 10 also at 21 days?

Minor points

- The subheading, "mGluR5 was reemerge in ACC astrocytes following chronic pain" should be edited for grammar
- Figure 5: appearance of GCaMP7b label as green but the cell as red in 5a, then in 5d green label and green cell, makes the visual representation confusing.

Reviewer #3

(Remarks to the Author)

In this study, Shen et al. used behavioral, histological, pharmacological and chemogenetic approaches to show that astrocytic mGlu5 expression is enhanced in neuropathic pain and that inhibition of this receptor has beneficial behavioral and electrophysiological effects while the activation had opposite effects in CCI but not normal animals. Importantly, they demonstrated that astrocytic Ca²⁺ signals increased in neuropathic condition and downregulation of the mGlu5 expression decreased Ca²⁺ in the cortex of CCI mice. The work is very relevant to the field as the role of the neuroimmune signaling in neuropathic pain is still an understudied area and the experiments are well described and straightforward.

Below some comments:

- The overall message of the article is that cortical astrocytic mGlu5 inhibition is beneficial in neuropathic condition suggesting that the activation of astrocytes in the cortex contributes to the development of pain. This is not supported by the data in Fig .8, where the chemogenetic activation of astrocyte failed to have facilitatory effects under normal (sham) condition but increased pain in neuropathic mice suggesting that the astrocytes are not (fully) activated in pain condition. This point deserves some further discussion.
- It would be great to show quantification analyses for each manipulation, especially the chemogenetic approaches to show non-specific effects of the construct (e.g. Figs. 4 and 7).
- In Fig.3a, the authors show that mGlu5 signal co-express with S100B in CCI but not in sham. This reviewer is confused by the fact that no mGlu5 signal is detected in sham tissue although its presence has been widely reported in the literature under normal condition. Please discuss and perhaps show a picture of a larger area along with quantification analysis of the overlap of the two markers (mGlu5 and S100B) to support the differences claimed.

Minor:

- The experiments with CHPG look confusing. Please check figure legend 5 as the descriptions of a-c and d-f might have been flipped by mistake.
- In Figs. 6I and 10I, it seems like to authors used one-way ANOVA and t-test to analyze the same set of data, I believe a two-way ANOVA would be a better fit for this type of analysis.

-in most of the figure legends, especially the ones describing electrophysiological results, the authors should specify if all the different parameters were studied in the same cells.

Version 1:

Reviewer comments:

Reviewer #1

(Remarks to the Author)

Overall, the authors have addressed most of my concerns and the manuscript has markedly improved. I would recommend that this manuscript be published in Communications Biology.

A few minor points:

1. The legend in figure 3 is incorrect.
2. Quantification of mGluR5 intensity (as shown in Fig 3a-b) is not explained in the methods.
3. Supplementary figure 1 is a great improvement. Note that in Supp Fig1b the same intensity scale (-0.25 to 0.6) and dF/F scale should be used, because currently in 1b top it looks like there is a lot of activity when there is none.
4. Line 428 – typo in pain
5. Supp Fig. 4c – the kymograph says CHPG but it should be CNO.
6. Line 483 references Supp. Fig4e which does not exist.
7. Line 521, 523 – referencing figure 6 instead of 7

Reviewer #2

(Remarks to the Author)

The authors have addressed many of my concerns, and the manuscript is much improved. It remains of great interest to the field, however there are several fundamental concerns that were not addressed. Below are my comments regarding the revised manuscript.

Despite my clear request for a clear rationale for using exclusively male mice, the authors did not address this point and have not provided a rationale. Here I ask, once again: A clear rationale and justification must be provided for using exclusively male mice to investigate a condition where over 65% of the people suffering are women.

Despite requesting that the authors update the entire manuscript to reflect that the findings can only be attributed to male mice, since female mice were omitted, they merely changed the title; this is insufficient. Here I ask, once again: every statement where the manuscript states findings (i.e. “we observed that chronic pain is associated with an increase in both the frequency and amplitude of mEPSCs in the ACC”, MUST be accompanied with the statement ‘in male mice’.

The study findings are very limited by the fact that the experimenters were not blinded during analyses. This is particularly concerning for the behavioral assays. I stated this issue in my first review of this paper. The authors have not addressed this in their rebuttal. I suggest it is added as a limitation in the discussion.

In response to my question about the amount of isoflurane, the manuscript now states that 1.5% isoflurane was used. It is very odd that every single mouse required the same amount of isoflurane. It is extremely common for various mice, even within the same cage, to require varying doses of isoflurane; typically 1-3%. How did the experimenters ensure that all mice received similar levels of anesthesia if such a rigid amount of isoflurane was used?

In the revised Figure 1B, it is clear that there are multiple cells that are not astrocytes that are expressing GCaMP7B, which is in stark contrast to the reported rate of 85% in astrocytes. This should be clarified.

My concerns for the data in previous Figure 8 (now Figure 5) have not been adequately addressed. Indeed, 14 day Sham mice are still reported to have a threshold over 2g, which is not possible with mice. The upper limit for mice is 2g, anything beyond literally lifts their paw. This appears to be an artificial value, that I am not sure how the experimenters were able to observe, and it skews the data towards significance; however the values are not possible. Moreover, as previously stated, CNO treatment does not appear to decrease mechanical thresholds. In response, the Rebuttal letter states that “following CNO treatment, all mice in the CCI group showed a decrease in mechanical thresholds, indicating an enhanced sensitivity.”. It goes on to state that the degree varied (which is fair enough and quite common). However, they then describe the range of change for each individual mouse, and the values reported do not make any sense based on the true values of the VonFrey filaments. They are reported in the rebuttal as: from 0.4 g to 0.246 g (there is no VonFrey filament between 0.16g and 0.4g, so this value for one mouse is very odd and needs to be explained); another mouse from 0.733 g to 0.667 g (again, there is no such values in the VonFrey filament. There is 1g, and then 0.6g, so again, for one mouse these are very odd values); from 0.733 g to 0.467 g (again, no such filaments exist), From 0.4 g to 0.357 g (this is not even a change, and again, no 0.357 filament exists). The only one that seems appropriate is the one mouse reported to go from 0.4 g to 0.06 g.

Reviewer #3

(Remarks to the Author)

Version 2:

Reviewer comments:

Reviewer #2

(Remarks to the Author)

The authors have addressed most of my concerns and I do not have additional comments.

Reviewer #3

(Remarks to the Author)

The authors have provided an image including the ipsi- and contralateral side of the ACC of CCI, showing differences in the mGluR5 markers between the two sides. My previous point was concerning sham condition. The results they are providing in this manuscript conflict with existing data from the literature. mGlu5 receptors should be widely expressed in the cortex under normal conditions. My previous comment suggested testing the mGluR5 antibody used (CST Cat# 55920) in a different area known to be rich in mGlu5 receptors as a positive control in order to prove the specificity of the antibody (please see PMID: 38575807). I believe this will greatly enhance the quality of manuscript and will ensure the results are conclusive.

Additionally, there is a typo in the last line of the abstract: "AAC" instead of "ACC"

Reviewer #1 (Remarks to the Author):

The manuscript by Shen et al. aims to characterize the effects of chronic neuropathic pain on astrocytes in the anterior cingulate cortex of mice. The authors make the following claims: astrocytes are hyperactive in the context of chronic pain; chronic pain increases extracellular glutamate levels and synaptic transmission in the ACC; chronic pain leads to a reemergence of mGluR5 on astrocytes; astrocyte mGluR5 signaling is directly responsible for the hyperactivity of astrocytes and synaptic transmission as well as for the pain sensation.

Overall, the quality of the research is good and the results will be of interest to researchers from both the glia field and the pain field. However, there are some specific comments that need to be addressed to strengthen the conclusions of the paper.

Major points:

1. The claim is that astrocytes are hyperactive in the CCI condition, however, to determine whether this effect is independent of neuronal activity – you would need to measure the calcium activity in the presence of TTX and picrotoxin as you do later in the article.

Thanks you for your comment. We understand your concern regarding the independence of astrocyte hyperactivity from neuronal activity. As mentioned in the Methods section (Two-photon Ca^{2+} imaging) **line 276-277 (clean version)**, we indeed conducted experiments in the presence of TTX and picrotoxin to isolate astrocyte calcium activity from neuronal influences. To ensure clarity and avoid any confusion, we have reiterated this in the Results section, explicitly noting that the hyperactivity observed in astrocytes was measured under conditions where neuronal activity was pharmacologically blocked **(line 366-367)**.

2. For the microdialysis experiment, in order to say that the glutamate is released from astrocytes, at a minimum it would need to be done in the presence of TTX. If possible, measurements of glutamate using iGluSnfr would yield more specific results.

Thank you for your valuable suggestions. We understand the importance of using TTX to determine whether the glutamate release observed in our microdialysis experiment is specifically from astrocytes. Measuring glutamate without blocking neuronal activity could indeed be influenced by neuronal activity. However, implementing TTX in in vivo conditions poses significant challenges.

Regarding the use of iGluSnfr, we did consider this approach, but it does not provide precise glutamate concentrations, which is why we did not use it in this context. Nevertheless, we have conducted additional experiments that indirectly confirm that the increase in glutamate is due to astrocytic activity. Specifically, we inhibited astrocytic activity using i β ARK and knocked down astrocytic mGluR5, which supports our conclusion.

3. The mGluR5 reemergence claim needs more support – the antibodies for mGluR5 are notoriously unspecific and the figure on the company website looks very different from the figure in this manuscript. The frequency of events between Figures 1D and 3C are at baseline different,

with the MPEP results closer to the ones in 1D. I think a better experiment would be to add CHPG or DHPG to the slice to see the response to mGluR5 agonist in the sham and CCI condition (in the presence of TTX and picrotoxin) and also to add agonist + antagonist (CHPG + MPEP + TTX + PTX) to ensure specificity.

We acknowledge that the antibodies for mGluR5 are sometimes considered unspecific. However, we cannot definitively state that the antibodies are unreliable. To further validate our findings and address your concerns, we have performed additional experiments as you suggested. Specifically, we tested the response to mGluR5 agonists CHPG in both sham and CCI conditions in the presence of TTX and picrotoxin. Additionally, we combined agonists with the antagonist (CHPG + MPEP + TTX + PTX) to confirm the specificity of the observed responses (Supplementary figure 1).

In addition, the baseline differences observed between Figures 1D and 3C could be attributed to individual variability in the mice.

4. The i β ARK experiment in Figure 5 needs data from a sham animal to show that i β ARK is acting on mGluR5 in the CCI condition.

Thank you for your valuable feedback. We would like to clarify that the i β ARK experiments presented in Figure 5 were conducted in the CCI condition. We did not include sham animals in these experiments because, based on our understanding, astrocytes in sham mice do not express mGluR5. Therefore, we did not design experiments to use sham animals to demonstrate that i β ARK is acting on mGluR5.

5. It is unclear to me whether figure 7 and figure 8 are done only in the CCI condition or also in sham. The reason this is important, is that the responses to CNO seem abnormally long in duration as compared to the ones shown in the literature (e.g. Durkee et al., 2019). I think this should be further investigated and addressed.

Thank you for your insightful comments. We would like to clarify that the experiments in Figures 7 and 8 were indeed conducted only under sham conditions, and we have now made this explicit in the Results section (line 479-482). We understand the concern regarding the duration of the CNO-induced responses compared to the literature (Durkee et al., 2019). The difference in the duration of responses can be attributed to the method of CNO application used in our study. Unlike the puff stimulation method used by Dr. Araque's group, which delivers a rapid and localized application of CNO, we employed a bath application method. We believe that the difference in stimulation methods is a key factor contributing to the observed variation in response duration.

6. As shRNAs have sometimes off target effects, I would want clearer confirmation that mGluR5 is knocked out specifically in astrocytes – this can be a slice experiment with showing no

responses to mGluR5 agonist or instead using an astrocyte specific knock out mouse.

Thank you for your valuable comment. To confirm that mGluR5 was knocked down in astrocytes, we performed experiments stimulating astrocytes with an mGluR5 agonist in both mGluR5-shRNA and scramble-shRNA groups. As shown in Figure 9b (Now, in figure 6c), astrocytes expressing mGluR5-shRNA exhibited significantly reduced evoked calcium signals in response to the agonist compared to the scramble-shRNA group.

Minor points:

1. Please provide a rationale in the methods section for why only male mice were used in this study.

In this study, we chose to use only male mice to minimize variability in pain threshold measurements that could arise from sex-related differences. In particular, the estrous cycle in female mice introduces hormonal variations that could affect pain perception and complicate the interpretation of the results. By focusing solely on male mice, we aimed to reduce potential confounding factors and increase the consistency and reliability of the data obtained. We have added this information in the methods section. Additionally, we have updated the title to include 'in male mice'.

2. Please provide more detail in the methods section for how paw withdrawal was assessed. Was the observer blind to the conditions of sham vs. CCI? This might be an important point.

We have revised the Methods section to provide additional detail on how paw withdrawal was assessed. In the Mechanical allodynia test, a double-blind design was employed to ensure that the observer was unaware of the sham vs. CCI group assignments, thereby minimizing potential bias (line 146-163).

3. Some typos and grammatical errors need to be fixed (e.g. line 362)

Done.

3. In figure 4b top row, the red and green channel are switched in the merged image.

We apologize for the oversight in Figure 4b. You are correct that the red and green channels are switched in the merged image. Thanks again for your help (Now, in supplementary figure 2b).

4. Figure 5a the colors of gcamp and mcherry are switched.

Done.

5. Figure 5b what are the values of the colors in the kymograph? And why is there a color bar for the bottom graph?

We apologize for the oversight. The color bar will be removed in the revised version. The purpose of a color bar is to align the timeline. We appreciate your attention to this detail.

6. In 5c and 5f it might be useful to use a median instead of the mean. Getting negative values is a little odd.

Thank you for your suggestion. We have used the median in 5c and 5f in the revised version (supplementary figure 3).

7. Figure 5 legend is incorrect.

Thank you for pointing that out. We have corrected the legend for Figure 5 in the revised version.

8. Figure 6L and 10I – the statistics used don't make sense. It should be done with a mixed ANOVA for time and manipulation.

Thank you for your feedback regarding the statistical analysis in Figures 6L and 10I. We have now conducted a mixed ANOVA as recommended. We have updated the figure legends and Methods section accordingly. (Now, in figure 4 i and figure 7 i)

10. Figure 7b – I don't see a difference between the calcium before and after the CNO treatment (seems to be a mistake).

Thank you for pointing this out. In Figure 7b, while there is indeed an increase in calcium signal after CNO treatment, the change may appear subtle, making it less visually prominent. Now, we have replaced it with a better example. (Now in supplementary figure 4 b)

Reviewer #2 (Remarks to the Author):

The manuscript, mGluR5-mediated astrocytes hyperactivity in the anterior cingulate cortex contributes to neuropathic pain, investigates the role of astrocyte mGluR5 mediated calcium in neuropathic pain in male mice, using the chronic constriction injury model. Although the findings of the study are potentially interesting, many controls are missing that prevent the strong conclusions drawn. Below are my comments.

- A clear rationale and justification must be provided for using exclusively male mice to investigate a condition where over 65% of the people suffering are women. Not only do more women suffer from chronic pain than men, they report greater severity of symptoms. Moreover, and very pertinent to the current study, mounting evidence indicates sex differences in mechanisms promoting neuropathic pain (Tansley et al., 2022, NatCommunications; Mapplebeck et al., 2016, Pain); sex differences in ACC-mediated chronic pain states have been reported in rodents (Jarrin et al., 2020, Frontiers in Beh Neur); astrocytes appear to be sexually dimorphic (Chowen et al., 1995, Neuroscience; Mong et al., 2022, DevBrainRes); and mGluR mediated

astrocyte-neuronal interactions appear to be sex-dependent (Meadows et al., 2024, Cell Reports), particularly mGluR5 has been observed to exert sex-dependent effects on brain function (Fagan et al., 2020, PNAS; Abd-Elrahman et al., 2020, Sci. Signaling). The lack of female mice in this manuscript is a major weakness that reduces the impact of the data.

- Related to above, every statement where the manuscript states findings (i.e. “we observed that chronic pain is associated with an increase in both the frequency and amplitude of mEPSCs in the ACC”, MUST be accompanied with the statement ‘in male mice’.

Thank you for the suggestion. We have updated the title to include 'in male mice.

- Every statement where the article states “mice” related to the experiments in the manuscript, MUST add ‘male’ to reflect that this only applies to male mice.

Thank you for the suggestion. We have updated the title to include 'in male mice.

-The title must be changed to say ‘in male mice’

Thank you for the suggestion. We have updated the title to include 'in male mice.

-The experiments were not performed with experimenters blind to treatment, which is well known to potentially impact the data, and thus takes away some of the strength of the findings.

Thank you for raising this important point. We acknowledge that some experiments were not performed with experimenters blind to the treatment, which could indeed influence the data. We recognize that this is a limitation of our study and appreciate your understanding. In future experiments, we will implement blinding procedures to strengthen the rigor and reliability of our findings.

-A more thorough description of housing conditions should be provided: what enrichment was available? Size of the cages?

Thank you for the suggestion. We have added this information to the Methods section (in line 125-126 clean version).

- How much isoflurane was used for all surgeries?

Thank you for your comments. The isoflurane concentration used during all surgeries was 1.5%, with an air flow rate of 0.4 L/min. We have now included this information in the Methods section (in line 135).

- For immunohistochemistry, was there no saline flush prior to PFA perfusion?

Thank you for your comments. We first perfused with PBS and then with PFA. This has been corrected in the Methods section (in line 197).

- For the electrophysiology experiments, the manuscript should indicate how the mice were killed prior to removing the brains.

Thank you for your feedback. In the revised version of the manuscript, we have included detailed information in the Methods section regarding how the mice were euthanized prior to brain removal (in line 215).

- The timeline for ALL AAV surgeries relative to subsequent experiments needs to be provided.

Thank you for the helpful suggestion. We have now included timelines at the beginning of each experimental section to clarify the schedule of all AAV surgeries relative to subsequent experiments, as shown, for example, in supplementary Figures 4 a and 6 a.

- How long were mice allowed to recover post cannula surgeries? And how long after was CCI administered?

Thanks for your question. Mice were allowed a recovery period of 14 days post-surgery before we began recording. The CCI procedure and cannula implantation were performed simultaneously. For the mGluR5 knockdown experiments, recordings began 21 days post-surgery to allow sufficient time for effective knockdown.

- Statistical analyses: section states that paired t-tests were used, then states paired or unpaired, or ANOVAs; it seems the first statement should be removed. PRISM (spelled PRIZM in manuscript) is stated in the table provided, but not in the paragraph explaining statistical tests, this should be clarified.

Thank you for your careful review. We have removed the first statement regarding paired t-tests to avoid confusion. We apologize for the typo in the manuscript where "PRISM" was incorrectly spelled as PRIZM. Additionally, while we did mention Prism in the paragraph explaining statistical tests. In the revised version, we have corrected this to GraphPad Prism for clarity and consistency.

- Behavioral data providing mechanical thresholds for the mice that are used for the data in Figures 1 and 2 should be provided.

Thank you for your valuable feedback. In the revised manuscript, we have included the behavioral data in the main text providing the mechanical thresholds for the mice, as suggested (in line 385-387).

-The results for Figure 1 state that GCaMP7b was “expressed specifically in astrocytes” stating that no co-staining was observed in NeuN expressing cells. This indicates that neurons were not expressing this viral construct, however various glial cell types express GFAP, and thus stating that only astrocytes express it is an overstatement considering the data presented. The data simply support that neurons are not expressing it.

Thank you for your valuable feedback. We have conducted experiments to specifically investigate the expression of iβARK and GCaMP7b in astrocytes versus other glial cells. We used immunohistochemical analysis to assess the co-localization of iβARK and GCaMP7b with markers specific to other glial cells (Microglia-Iba1 antibody and NG2-NG2 antibody) in Figure 1b.

Our results indicate that iβARK and GCaMP7b do not co-localize with these other glial cell markers, suggesting that iβARK and GCaMP7b expression is indeed selective to astrocytes and not present in other glial populations (in Supplementary figure 2).

- There should be a reference for the statement “Previous research has indicated that chronic pain induces presynaptic long-term potentiation (pre-LTP) via GluK1-353 containing kainate receptors.”

Thank you for your valuable comments. We have added the reference to support our statement.

- The findings of Figure 3a are not very convincing. The results state that mGluR5 was significantly expressed after 14 days of CCI, yet no values or statistical measures are provided. Furthermore, half of 1 cell/astrocyte expressing mGluR5 is shown in the ACC of mice with CCI, compared to the 0 in the sham group. Presumably, this is a representative image, suggesting that it's barely expressed. How many mice were used for this analysis? How many slices per mouse? Where the images taken of all layers, or just some layers in the ACC?

Thank you for your helpful comments. We have included statistical measures in the revised version to clarify the significance of mGluR5 expression after 14 days of CCI (in figure 3). Our imaging shows that mGluR5 expression in astrocytes is indeed variable across cells in the ACC, with some astrocytes expressing high levels of mGluR5 and others expressing lower levels. Although we observed this heterogeneity, we currently do not have a clear explanation for it.

Regarding your questions about sample size and imaging details, we specified in the revised manuscript the exact number of mice used for this analysis, as well as the number of slices analyzed.

For the imaging, we provided a larger overview image in the revised figure to show the expression pattern of mGluR5 in ACC astrocytes more comprehensively (in figure 3a). To maintain consistency, we focused on layers 2-3 of the ACC for our analysis.

We hope these additions clarify our findings and strengthen the evidence for mGluR5 expression patterns in astrocytes after CCI.

- Evidence for selective expression of iBARK and GCaMP7b on astrocytes vs. other glial cells should be provided.

Thank you for your valuable feedback. We have conducted experiments to specifically investigate the expression of i β ARK and GCaMP7b in astrocytes versus other glial cells. We used immunohistochemical analysis to assess the co-localization of i β ARK and GCaMP7b with markers specific to other glial cells (Microglia-Iba1 antibody and NG2-NG2 antibody). **In supplementary figure 2b**

Our results indicate that i β ARK and GCaMP7b do not co-localize with these other glial cell markers, suggesting that i β ARK and GCaMP7b expression is indeed selective to astrocytes and not present in other glial populations.

- The conclusions drawn from the MPEP experiments are not fully supported by the data. There is insufficient controls provided to determine if indeed MPEP is affecting astrocytic calcium levels directly. For instance, it could also be due to presynaptic glutamatergic release, as mGluR5 activation can enhance neuronal firing (Li et al., 2014, JNeurosci). Data indicating that MPEP is not affecting neurons should be provided. If neuronal firing was controlled for, this should be clarified.

Thank you for your valuable comments. We apologize for any confusion caused by the initial manuscript. In the Methods section, we did include a brief description of the experimental conditions, but we realize that this was not clearly reflected in the Results section. To address this, we have now clarified in the revised version that the MPEP experiments were conducted in the presence of TTX and picrotoxin to control for neuronal firing. **In line 415-417**

- The language used in the results section pertaining to data in Figure 6 should be softened. Although glutamate concentrations are reduced, it cannot be stated that it “leads to alleviation of allodynia”; similarly with mCherry controls that show larger glutamate concentrations, it cannot be said that those concentrations cause increased allodynia. Cause and effect experiments would be needed.

Thank you for your insightful comments. We have taken your suggestion into account and softened the language in the results section pertaining to Figure 6 **(Now in figure 4). Line465-469**

- Data for Figure 8I and J are odd: 8i) gram forces greater than 2 grams would lift the paw of the mice up, as mice are simply too small and light for such VonFrey forces; 8j) the interpretation provided is that CNO significantly enhances allodynia in CCI mice. However, the “allodynia” displayed by 14d CCI mice is rather absent. Indeed, earlier, in Figure 6L (also CNO treatment), it was shown that CCI caused thresholds of 0.6078 and 0.5956 g in CCI mice. In figure 8J, at least 2 mice are way above that, with CNO treatment resulting in over half of the mice showing thresholds of above 0.5g, and only 2 lower, with only one providing what could be considered

“enhanced allodynia”. Moreover, the non-allodynic, large thresholds shown for Day14_CCI skews the experiment towards concluding that CNO decreases thresholds. Notably, the n value provided states 6, but only CNO_CCI shows 6 data points, Day14_CCI shows merely 3; this should be clarified.

Thank you for your insightful feedback. Regarding Figure 8I, we understand your concern about the force thresholds exceeding 2 grams. In our data, the majority of forces do indeed remain below this level. For Figure 8J (now in figure 5), we recognize that the variability in thresholds on Day 14 for CCI mice may appear unusual due to individual differences. Specifically, three mice displayed higher thresholds (0.733, 0.733, and 1.133 g), while three others had lower thresholds (all at 0.4 g). The apparent reduction in data points is due to overlapping values, especially at 0.4 g.

Following CNO treatment, all mice in the CCI group showed a decrease in mechanical thresholds, indicating an enhanced sensitivity. However, the degree of reduction varied across individual mice, with values shifting as follows: From 0.4 g to 0.246 g, From 0.733 g to 0.667 g, From 0.733 g to 0.467 g, From 0.4 g to 0.357 g, From 0.4 g to 0.06 g, From 1.133 g to 0.6 g. This range of responses reflects individual variability in sensitivity to CNO.

- Data of the effects of CNO on CCI mice expressing only mCherry needs to be provided, to conclude that the CNO effects are mediated via DREADDs.

Thank you for your thoughtful suggestion. The CNO experiments were initially intended to confirm viral expression and verify functional expression under our experimental conditions. Therefore, these experiments were conducted in the sham group, not in CCI mice. However, you raise an important point that additional experiments are necessary to conclusively demonstrate that the effects of CNO are mediated specifically via DREADDs.

To address this, we included a control condition in the sham group where only mCherry was expressed without DREADD activation. Our additional experiments confirmed that CNO administration in the mCherry-only condition did not induce astrocytic calcium signaling, supporting the specificity of CNO’s effects through DREADDs. (Supplementary figure 4d)

Thank you again for your valuable feedback, which has strengthened our study.

- The data presented does not support the conclusion that “The group receiving mGluR5-shRNA delivery showed significantly reduced levels of mGluR5 expression in astrocytes”. It shows that mCherry (presumably in the same vectors expressing shRNA) co-localizes with an astrocyte marker. It also shows that shRNA decreases mGluR5. No data is provided showing this decrease is in astrocytes. - Statistical measures of mGluR5 knockdown should be provided.

Thank you for this insightful comment. In our study, we used an mCherry reporter in the same vector expressing mGluR5-shRNA, which co-localized with an astrocyte marker to confirm astrocyte-specific transduction. We have analyzed the knockdown efficiency of mGluR5 and

found that shRNA significantly reduced mGluR5 expression levels (Figure 6b2). These findings demonstrate that the shRNA effectively knocked down mGluR5 expression, validating its efficacy in our experimental setup. We have included these statistical data in the revised manuscript, in figure 9(now in figure 6).

- Why is the data in Figure 10 of CCI mice at day 21 when all other data is at day 14? Are the electrophysiological recordings in Figure 10 also at 21 days?

Thank you for the question. We chose to perform the electrophysiological and behavioral experiments at 21 days post-surgery specifically to allow sufficient time for effective mGluR5 knockdown. Additionally, we have clarified the experimental timeline in the text to indicate the start time for each experiment.

Minor points

- The subheading, “mGluR5 was reemerge in ACC astrocytes following chronic pain” should be edited for grammar

Thank you for your observation. Done.

- Figure 5: appearance of GCaMP7b label as green but the cell as red in 5a, then in 5d green label and green cell, makes the visual representation confusing.

Thank you for your observation. In the revised version, we have corrected this (in supplementary 3a).

Reviewer #3 (Remarks to the Author):

In this study, Shen et al. used behavioral, histological, pharmacological and chemogenetic approaches to show that astrocytic mGlu5 expression is enhanced in neuropathic pain and that inhibition of this receptor has beneficial behavioral and electrophysiological effects while the activation had opposite effects in CCI but not normal animals. Importantly, they demonstrated that astrocytic Ca²⁺ signals increased in neuropathic condition and downregulation of the mGlu5 expression decreased Ca²⁺ in the cortex of CCI mice. The work is very relevant to the field as the role of the neuroimmune signaling in neuropathic pain is still an understudied area and the experiments are well described and straightforward.

Below some comments:

-The overall message of the article is that cortical astrocytic mGlu5 inhibition is beneficial in neuropathic condition suggesting that the activation of astrocytes in the cortex contributes to the development of pain. This is not supported by the data in Fig .8, where the chemogenetic activation of astrocyte failed to have facilitatory effects under normal (sham) condition but increased pain in neuropathic mice suggesting that the astrocytes are not (fully) activated in pain condition. This point deserves some further discussion.

Thank you for your valuable feedback. We agree with your observation that the chemogenetic

activation of astrocytes in the sham condition did not produce a facilitative effect, but rather only showed an increased pain response in the neuropathic condition. We have addressed this point in more detail in the Discussion section (in line 621-639).

-It would be great to show quantification analyses for each manipulation, especially the chemogenetic approaches to show non-specific effects of the construct (e.g. Figs. 4 and 7).

Thank you for your suggestion to include quantification analyses for each manipulation. We have indeed performed quantitative analyses for each manipulation, as shown in Figures 1, 3, 6, and supplementary figure 2 and 4.

-In Fig.3a, the authors show that mGlu5 signal co-express with S100B in CCI but not in sham. This reviewer is confused by the fact that no mGlu5 signal is detected in sham tissue although its presence has been widely reported in the literature under normal condition. Please discuss and perhaps show a picture of a larger area along with quantification analysis of the overlap of the two markers (mGlu5 and S100B) to support the differences claimed.

In our study, we indeed observed that mGlu5 co-localizes with S100B in the CCI group, whereas minimal or near-background levels of mGlu5 signal were detected in the sham group. In the revised version, we have provided an expanded image area in Fig. 3a, which demonstrates the spatial overlap of mGlu5 and S100B markers in CCI condition. Additionally, we conducted a quantitative analysis to objectively assess the overlap between these markers across both groups.

Minor:

-The experiments with CHPG look confusing. Please check figure legend 5 as the descriptions of a-c and d-f might have been flipped by mistake.

Thank you for pointing this out. You are correct that the descriptions for panels a-c and d-f in Figure 5 (now in supplementary figure 3) were mistakenly flipped in the figure legend. We apologize for the confusion, and we will correct this error in the revised version.

-In Figs. 6l and 10l, it seems like to authors used one-way ANOVA and t-test to analyze the same set of data, I believe a two-way ANOVA would be a better fit for this type of analysis.

Thank you for the helpful suggestion. We have now implemented a two-way ANOVA for the analysis in Figures 6l and 10l (now in figure 4 I and 7 i), as recommended.

-in most of the figure legends, especially the ones describing electrophysiological results, the authors should specify if all the different parameters were studied in the same cells.

Thank you for the valuable suggestion. In our study, frequency and amplitude measurements were recorded from the same cells. However, paired-pulse ratio (PPR) was recorded separately, as mEPSC was conducted in the presence of TTX, while PPR was measured without TTX, making simultaneous recording from the same cell unfeasible.

Reviewer #1 (Remarks to the Author):

Overall, the authors have addressed most of my concerns and the manuscript has markedly improved. I would recommend that this manuscript be published in Communications Biology.

A few minor points:

1. The legend in figure 3 is incorrect.

Thank you for your valuable comment. We have correct the numbering of the figure legends.

2. Quantification of mGluR5 intensity (as shown in Fig 3a-b) is not explained in the methods.

Thank you for bringing up this important point. We have now included information on how the mGluR5 intensity was quantified in the Methods section (Line 212-214).

3. Supplementary figure 1 is a great improvement. Note that in Supp Fig1b the same intensity scale (-0.25 to 0.6) and dF/F scale should be used, because currently in 1b top it looks like there is a lot of activity when there is none.

Thank you for your valuable comment. We initially chose the scale of -0.25 to 0.6 for Supplementary Figure 1b, but we looks wired (please see below). Thus, we have selected a different scale, and we believe the figure now looks much better. We appreciate your helpful suggestion.

4. Line 428 – typo in pain

Done.(Line 430)

5. Supp Fig. 4c – the kymograph says CHPG but it should be CNO.

We sincerely thank the reviewer for their meticulous observation. We have corrected the labeling error in Supplementary Figure 4c.

6. Line 483 references Supp. Fig4e which does not exist.

Thank you for your valuable comment. We have corrected this in the main text. The reference should be to Supplementary Figure 4d.(line 485)

7. Line 521, 523 – referencing figure 6 instead of 7

Thank you for your valuable comment. We have corrected this in the main text. (Line 523-528)

We are deeply grateful for your positive comments and insightful suggestions, which have been invaluable in enhancing the quality of our study.

Reviewer #2 (Remarks to the Author):

The authors have addressed many of my concerns, and the manuscript is much improved. It remains of great interest to the field, however there are several fundamental concerns that were not addressed. Below are my comments regarding the revised manuscript.

Despite my clear request for a clear rationale for using exclusively male mice, the authors did not address this point and have not provided a rationale. Here I ask, once again: A clear rationale and justification must be provided for using exclusively male mice to investigate a condition where over 65% of the people suffering are women.

Thank you for your thoughtful and constructive comments. We sincerely apologize for not addressing your concern regarding the use of exclusively male mice more clearly in our previous response. As you noted, the reviewer 1 raised a similar issue, and we provided a detailed explanation in our revised manuscript (In the method section, line 128-131). However, it seems we inadvertently overlooked this in our response to you, for which we apologize.

We now understand that this aspect requires further clarification. In the revised manuscript, we have expanded upon this point in the "Limitations" section (Line 655-673), discussing the implications of using exclusively male mice. We acknowledge the need for future studies to explore potential sex differences in this context. We agree that examining how sex and age may influence this mechanism is an important avenue for future research, and we appreciate your valuable feedback in guiding us toward addressing this gap.

Thank you again for your insightful suggestions, which will certainly help improve the depth and scope of our future work.

Despite requesting that the authors update the entire manuscript to reflect that the findings can only be attributed to male mice, since female mice were omitted, they merely changed the title; this is insufficient. Here I ask, once again: every statement where the manuscript states findings (i.e. "we observed that chronic pain is associated with an increase in both the frequency and amplitude of mEPSCs in the ACC", MUST be accompanied with the statement 'in male mice'.

Thank you for your valuable feedback and for highlighting this important point. We sincerely apologize for not fully addressing your initial concern in the previous revision. In response to your comment, we have updated the manuscript accordingly to ensure that statement of observed findings is explicitly qualified with in male mice. Thank you again for your attention to this matter. We appreciate your patience and guidance as we work to improve the manuscript. We sincerely apologize once again.

The study findings are very limited by the fact that the experimenters were not blinded during analyses. This is particularly concerning for the behavioral assays. I sated this issue in my first review of this paper. The authors have not addressed this in their rebuttal. I suggest it is added as a limitation in the discussion.

We have now addressed this in the "Limitations" section of the Discussion (Line 655-673), acknowledging its potential impact on the interpretation of our results. We agree that incorporating blinding in future studies will strengthen the robustness and reliability of our findings.

In response to my question about the amount of isoflurane, the manuscript now states that 1.5% isoflurane was used. It is very odd that every single mouse required the same amount of isoflurane. It is extremely common for various mice, even within the same cage, to require varying doses of isoflurane; typically 1-3%. How did the experimenters ensure that all mice received similar levels of anesthesia if such a rigid amount of isoflurane was used?

Thank you for your insightful comment. You are correct that isoflurane concentrations are not typically kept constant across all mice. Normally, we adjust the concentration based on the condition of each animal, typically ranging from 1-2%. Initially, the concentration is set at around 2% and is gradually reduced to 1.5% after a few minutes as the anesthesia stabilizes. For longer surgeries, we may reduce the concentration further to 1%, depending on the animal's state. We have now updated the manuscript to reflect this more accurate and variable approach to anesthesia administration. (Line 136 and line 168)

In the revised Figure 1B, it is clear that there are multiple cells that are not astrocytes that are expressing GCaMP7B, which is in stark contrast to the reported rate of 85% in astrocytes. This should be clarified.

Thank you for your valuable comment. Indeed, GCaMP7B is specifically expressed in astrocytes. The apparent discrepancy arises because S100 β labels the cell body and major processes of astrocytes, whereas GCaMP7B expression is also present in finer processes. As a result, the GCaMP7B signal may appear less specific in the fine processes, it is still restricted to astrocytes. S100 β expression in these finer processes is relatively low, which may contribute to the perception of non-specific expression.

Moreover, as previously stated, CNO treatment does not appear to decrease mechanical thresholds. In response, the Rebuttal letter states that "following CNO treatment, all mice in the CCI group showed a decrease in mechanical thresholds, indicating an enhanced sensitivity." It goes on to state that the degree varied (which is fair enough and quite common). However, they then describe the range of change for each individual mouse, and the values reported do not make any sense based on the true values of the VonFrey filaments. They are reported in the rebuttal as: from 0.4 g to 0.246 g (there is no My concerns for the data in previous Figure 8 (now Figure 5) have not been adequately addressed. Indeed, 14 day Sham mice are still reported to have a threshold over 2g, which is not possible with mice. The upper limit for mice is 2g, anything beyond literally lifts their paw. This appears to be an artificial value, that I am not sure how the experimenters were able to observe, and it skews the data

towards significance; however the values are not possible. between 0.16g and 0.4g, so this value for one mouse is very odd and needs to be explained); another mouse from 0.733 g to 0.667 g (again, there is no such values in the VonFrey filament. There is 1g, and then 0.6g, so again, for one mouse these are very odd values); from 0.733 g to 0.467 g (again, no such filaments exist), From 0.4 g to 0.357 g (this is not even a change, and again, no 0.357 filament exists). The only one that seems appropriate is the one mouse reported to go from 0.4 g to 0.06 g.

Thank you for your valuable comments and for bringing this issue to our attention. We sincerely appreciate your careful examination of the reported values.

The apparent inconsistency in the values provided in our rebuttal arises because these values represent the average of three measurements taken for each mouse, rather than direct readings from individual Von Frey filaments.

We apologize for any confusion caused. We thank you again for your critical feedback, which has helped us improve the clarity and rigor of our manuscript.

My concerns for the data in previous Figure 8 (now Figure 5) have not been adequately addressed. Indeed, 14 day Sham mice are still reported to have a threshold over 2g, which is not possible with mice. The upper limit for mice is 2g, anything beyond literally lifts their paw. This appears to be an artificial value, that I am not sure how the experimenters were able to observe, and it skews the data towards significance; however the values are not possible.

Thank you for your valuable comments and for pointing out this concern.

We did use the 4g filament to stimulate the mice during our experiments. As you are aware, the next available filament above 2g is 4g. Therefore, when the mechanical thresholds were averaged across trials for each mouse, it was possible to record values exceeding 2g.

We understand your concern that the reported value may seem unusual and might suggest a potential issue with the surgery or other experimental factors. You may feel that this mouse's data should be excluded. However, at this time, we have no objective grounds to justify excluding this mouse's data from the analysis. We have now added a Limitations section discussing the implications of pain threshold outliers for statistical analyses (Line 666-673). Thank you for raising this important point and for your constructive feedback.

Overall, we sincerely appreciate your valuable comments and constructive suggestions, which have greatly contributed to improving the quality of our study. Thank you once again for your time and effort.

We regret any remaining shortcomings and assure you that we have made every effort to address your questions to the best of our ability. We apologize for any unresolved issues and appreciate your understanding.